# PAWS: Preference Learning with Advantage-Weighted Segments

**Aleksandar Taranovic** [1]   **Onur Celik** [1]   **Niklas Freymuth** [1]   **Ge Li** [1]   **Serge Thilges** [1]   **Huy Le** [1,2]   **Tai Hoang** [1]
**Rania Rayyes** [3]   **Gerhard Neumann** [1]

## Abstract

Preference-based reinforcement learning (PbRL) learns policies from human trajectory-level comparisons, avoiding explicit reward design and expert demonstrations. Existing methods typically train utility functions on trajectory or segment-level preferences while relying on per-step utility estimates during policy optimization. This training and inference mismatch induces a distribution shift that severely degrades temporal credit assignment and limits policy learning. We analyze this issue and propose PAWS[*], a segment-based preference learning method that performs policy updates directly using segment-level advantage functions. By aligning utility training with policy optimization, PAWS preserves trajectory-level preference information and avoids unreliable per-step learning signals. Experiments on simulated robotic manipulation and locomotion tasks demonstrate that PAWS consistently outperforms existing PbRL approaches, highlighting the importance of distribution-consistent preference learning.

## 1. Introduction

Preference-based reinforcement learning (PbRL) aims to learn policies that align with human preferences by leveraging pairwise comparisons between behaviors rather than explicit reward functions (Wirth et al., 2017; Christiano et al., 2017). In this setting, a human evaluator is presented with two candidate behaviors and provides a binary label

[1]Autonomous Learning Robots, Karlsruhe Institute of Technology, Karlsruhe, Germany [2]Bosch Center for Artificial Intelligence, Renningen, Germany [3]Institute for Material Handling and Logistics (IFL), Karlsruhe Institute of Technology, Karlsruhe, Germany. Correspondence to: Aleksandar Taranovic <aleksandar.taranovic@kit.edu>.

*Proceedings of the $43^{rd}$ International Conference on Machine Learning*, Seoul, South Korea. PMLR 306, 2026. Copyright 2026 by the author(s).

[*]The project webpage with code: https://ataranovic.github.io/PAWS-webpage/

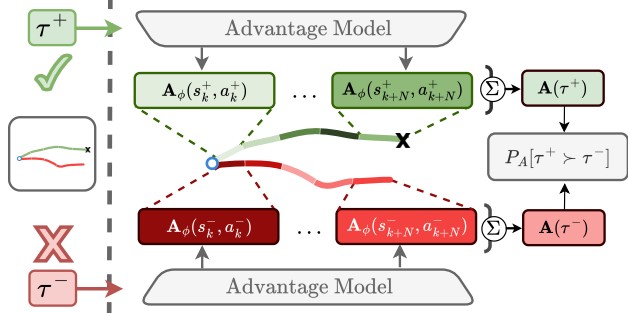

*Figure 1.* **The Temporal Credit Assignment Problem.** Illustration of learning a trajectory-level advantage value $A(\tau)$ from preference data. The advantage model is trained on preferred ($\tau^+$) and non-preferred ($\tau^-$) trajectories using the loss $P_{A_\phi}[\tau^+ \succ \tau^-]$. This loss depends only on the sum $A_\phi(\tau) = \sum_k A_\phi(s_k, a_k)$ of per-step advantages, which constrains the model at the trajectory level. As a result, many different assignments of individual state-action advantages are consistent with the same preference label (Figure 2). During policy optimization, however, the advantage function is queried on individual state-action pairs (Christiano et al., 2017; Kim et al., 2023), inducing a distribution shift between segment-level training and step-level inference.

indicating which one is preferred under implicitly defined criteria. The resulting feedback is often easier and more reliable to provide than demonstrations or manually specified rewards, especially for complex or subjective tasks. This is particularly true when demonstration quality varies widely, e.g., teleoperation scenarios with operators of differing expertise.

In sequential decision-making problems such as robotic control, preferences are typically expressed over trajectory segments, which are sequences of states or state-action pairs, rather than individual steps. Consequently, most PbRL methods train a utility function, such as a reward or advantage model, using segment-level or trajectory-level comparisons. Policy optimization, however, is usually performed at the level of individual state-action pairs, which requires the learned utility to be inferred on single steps.

The mismatch between segment-level training and step-level inference induces a distribution shift that is largely overlooked in existing PbRL approaches. Figure 1 illustrates how a utility model, in this case the advantage function,

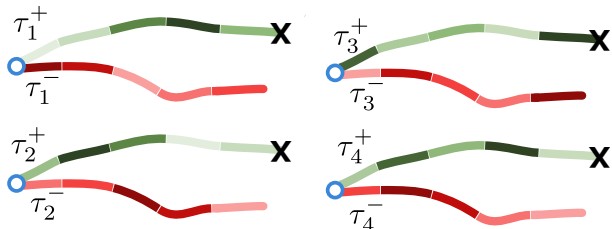

*Figure 2.* **Ambiguity in Per-Step Credit Assignment.** Four pairs of preferred ($\tau^+$) and non-preferred ($\tau^-$) segments are shown, with the intensity of green and red encoding the per-step advantage. The sum of advantages within each segment is identical across the two pairs, so they yield the same trajectory-level preference label, yet the per-step assignments differ markedly. This illustrates the underdetermined nature of temporal credit assignment under the distribution shift between segment-level training and step-level inference, where any consistent per-step assignment is equally compatible with the supervision signal, leaving downstream policy updates exposed to arbitrary choices made by the utility model.

is trained. Although the utility model is trained to accurately rank complete segments, it is later used to evaluate individual state-action pairs. For example, in Figure 2, we show four possible per-step utility distributions for the preferred and non-preferred segments. Here, a segment $\tau_i^{+/-}$ consists of five steps, with the utility of each step encoded by color intensity, where darker green/red shades indicate higher/lower utility values. The sum of utility of all steps in the segment is the same, but the utility of individual steps can vary significantly. Thus, many distinct per-step utility assignments can explain the same segment-level preference, leaving temporal credit assignment fundamentally ambiguous. This ambiguity severely limits the ability of the utility model to provide meaningful learning signals during policy optimization, especially when the preferences span a wide range of behavior quality.

This issue is commonly referred to as the temporal credit assignment problem in preference learning (Wirth et al., 2017). Crucially, we argue that this problem does not merely arise from the absence of fine-grained labels, but from a distributional inconsistency between how utilities are trained and how they are used. As we empirically demonstrate, when the same learned utility is queried with segment-level inputs that match its training distribution, prediction quality improves substantially, and preference information becomes significantly more informative for temporal credit assignment.

To address this mismatch, we introduce Preference Learning with Advantage-Weighted Segments (PAWS). PAWS performs policy optimization directly at the segment level by using a learned advantage function that is trained and queried consistently on trajectory segments. By combining segment-wise advantage estimation with a trust-region-constrained update, PAWS preserves trajectory-level preference information while avoiding unreliable single-step credit assignment.

This enables stable and effective policy updates that remain close to the data distribution implied by the preferences. This formulation further enables a principled, data-driven control of policy update strength through the effective sample size of advantage-weighted segments, reducing reliance on manually tuned optimization hyperparameters.

Our contributions are fourfold. First, we analyze temporal credit assignment in PbRL through the lens of training and inference distribution shift, identifying it as a core limitation of existing methods. Second, we propose PAWS, a segment-based preference learning approach that aligns utility training with policy optimization, thereby enabling more reliable propagation of preference signals. Third, we introduce an intuitive, data-driven strategy for setting policy optimization hyperparameters based on the effective sample size of preference-weighted data. Finally, we validate our method on a diverse set of simulated robotic manipulation and locomotion tasks, demonstrating consistent performance improvements over established baselines.

## 2. Related Work

**Preference-based Reinforcement Learning (PbRL).** PbRL leverages comparative feedback as the primary supervision signal (Wirth et al., 2017). Some approaches integrate entropy-based regularization (Lee et al., 2021a) or behavior cloning pretraining, where models are initialized using demonstrations of desired behavior. In LLM fine-tuning, a KL-divergence term is frequently introduced to anchor the trained policy to its pretrained counterpart (Ziegler et al., 2019; Stiennon et al., 2020; Ouyang et al., 2022; Guo et al., 2025). Online preference learning methods, such as those proposed by Verma & Metcalf (2024), tackle the issue of credit assignment. However, they rely on online interaction and require substantially more data. Gao et al. (2024) propose a similar offline preference learning approach, but it depends on massive amounts of unlabeled data.

**Direct preference learning.** Recent methods such as DPO (Rafailov et al., 2023), CPL (Hejna et al., 2024), IPL (Hejna & Sadigh, 2023), DPPO (An et al., 2023), and PPL (Cho et al., 2025) implicitly avoid the reward assignment problem. Instead of learning a reward function, they explicitly optimize the policy likelihood from preferences. However, these methods do not utilize reward models to capture complex dependencies between different state-action pairs. As such, they struggle on tasks that require complex reasoning (Ivison et al., 2024; Xu et al., 2024) and in settings of limited preference data.

**Feedback Collection.** The process of acquiring preference data varies significantly across methods. Certain frameworks gather feedback online, collecting trajectories or comparisons dynamically during training (Lee et al., 2021a;

Christiano et al., 2017), while others rely on static, pre-collected datasets (Hejna et al., 2024). Preference query generation also varies between methods (Lee et al., 2021b). Our work focuses on offline datasets, though PAWS is agnostic to the data collection process and can also be applied to online settings. A critical consideration in feedback collection is the quality of preference labels (Lee et al., 2021b). Many approaches address noisy or conflicting data by aggregating signals from multiple policies or annotators. We focus on informative preferences, where an oracle with access to expert metrics, such as ground-truth rewards or Q-functions, provides labels. Combining preferences with other feedback modalities, such as demonstrations, is an open challenge that has been tackled in several recent works (Biyik et al., 2021; Taranovic et al., 2023; Ibarz et al., 2018). Similarly, several recent works consider alternative feedback types (Abdolmaleki et al., 2025; Myers et al., 2021).

**Reward Function Learning.** The goal in PbRL is to train a reward (Ng et al., 2000) or a utility-based value function model (Knox et al., 2024) that aligns with the collected preferences. The Bradley–Terry (BT) model (Bradley & Terry, 1952) serves as a foundational approach for many methods. Here, human preference is modeled as a Boltzmann rational distribution (Baker et al., 2009) over the sum of the discounted rewards of individual time steps. However, recent research suggests that human preferences are better captured through regret-based formulations (Knox et al., 2024) where the advantage of the optimal policy is used to model the quality of a segment instead of the rewards. Most BT-inspired methods optimize log-likelihood objectives, or use logistic separation and hinge loss variants (Taranovic et al., 2023). To better reflect the complexity of human preference structures, reward models are often modeled with a Transformer-based architecture (Nakano et al., 2021; Gao et al., 2023; Zhao et al., 2024) capable of learning non-Markovian rewards (Kim et al., 2023). When feedback is collected online, the reward model must be continually updated to incorporate new data (Taranovic et al., 2023; Lee et al., 2021a).

**Policy Optimization.** The choice of the RL method for policy optimization often depends on the data collection paradigm. For online settings, trust-region-constrained on-policy approaches (Schulman et al., 2015; 2017; Abdolmaleki et al., 2018; Li et al., 2024; Otto et al., 2021; Hoang et al., 2025) are commonly used, while off-policy algorithms (Haarnoja et al., 2018) are typically combined with relabeled rewards (Lee et al., 2021a). In offline scenarios, algorithms like IQL (Kostrikov et al., 2021) are frequently adopted to handle the constraints of static datasets. In the reward-weighted regression setup (Peters & Schaal, 2007; Neumann & Peters, 2008) and related Expectation Maximization (EM)-like approaches (Abdolmaleki et al.,

2018), the constrained policy optimization usually leads to a weighted maximum likelihood optimization that avoids querying the credit assignment out of distribution (Kostrikov et al., 2021; Nair et al., 2020). Our method diverges from conventional PbRL methods by directly learning the advantage function, removing the need for separate reward training. Additionally, we propose updating the policy on segments instead of single state-action pairs, similar to some policy optimization methods in episodic RL (Kober & Peters, 2008; Abdolmaleki et al., 2015; Daniel et al., 2016; Li et al., 2024; 2025). We base the policy update scheme on the EM-like, trust-region-constrained optimization problem (Peters et al., 2010), but adapt it to the offline setting.

## 3. Method

We present PAWS, a preference learning method designed to address the distribution shift between utility training and policy optimization in preference-based reinforcement learning. Given previously collected preference data over trajectory segments, we first train an advantage function $A_\phi$. We then optimize the policy by performing updates directly on trajectory segments, using advantage-weighted segment returns rather than per-step utility estimates.

**Notation.** We consider a policy $\pi_\theta : \mathcal{S} \times \mathcal{A} \to \mathbb{R}^+$ parameterized by $\theta$, where $\mathcal{S}$ and $\mathcal{A}$ denote the state and action spaces. The policy induces a distribution over actions conditioned on states. Preference learning is performed by training an advantage function $A_\phi : \mathcal{S} \times \mathcal{A} \to \mathbb{R}$ with parameters $\phi$, which assigns a scalar utility to state-action pairs. We assume access to trajectories $\tau_T^i = (s_0^i, a_0^i, \ldots, s_{T-1}^i, a_{T-1}^i)$ of length $T$. Since human preferences are hard to elicit reliably over long trajectories, we operate on fixed-length trajectory segments, which are short enough for consistent comparison while still capturing temporally extended behavior. A segment of length $N$ starting at time step $k$ is denoted as $\tau_{k:N}^i = (s_k^i, a_k^i, \ldots, s_{k+N-1}^i, a_{k+N-1}^i)$. Segments are sampled from an unknown data distribution $p_D(\tau)$, from which we assume access to $K$ segment samples. When temporal indexing is not essential, we denote a generic segment simply as $\tau_i$. Preference data consist of pairs of trajectory segments, where $\tau_i^+$ is preferred over $\tau_i^-$. The preference dataset is denoted as $D_{\text{pref}} = \{(\tau_i^+, \tau_i^-)\}_{i=1}^n$, and we use $\succ$ to denote the preference relation.

### 3.1. Advantage Learning

Following Knox et al. (2024), we model human preferences using an advantage function rather than partial returns, which better reflects how humans evaluate temporally extended behavior. The advantage function is trained on segment-level comparisons, and the likelihood of preferring

segment $\tau^+$ over $\tau^-$ is defined as

$$P_{A_\phi}[\tau^+ \succ \tau^-] = \frac{\exp\left(A_\phi(\tau^+)\right)}{\exp\left(A_\phi(\tau^+)\right) + \exp\left(A_\phi(\tau^-)\right)}, \quad (1)$$

where $A_\phi(\tau) = \sum_t A_\phi(s_t, a_t)$. The advantage values of both segments are aggregated, and training reduces to binary cross-entropy optimization over the difference in cumulative advantage predictions. The resulting loss is

$$\mathcal{L}_{\text{pref}}(\phi) = -\frac{1}{n}\sum_{i=1}^n \log \sigma\left(A_\phi(\tau_i^+) - A_\phi(\tau_i^-)\right). \quad (2)$$

Importantly, this objective constrains the advantage function only at the segment level. While $A_\phi$ is trained to correctly rank trajectory segments, many different per-step advantage assignments can yield identical segment-level sums, as shown in Figure 2. Thus, the learned advantage function is underconstrained at the level of individual state-action pairs. Most existing PbRL methods subsequently query this advantage or reward model at the per-step level during policy optimization, thereby inducing a distribution shift between the segment-level training and the step-level inference. This mismatch leads to ambiguous and unreliable temporal credit assignment. In contrast, PAWS preserves distributional consistency by using the learned advantage function directly on trajectory segments during policy optimization.

We consider two architectures for $A_\phi$: an encoder-only Transformer (Vaswani et al., 2017) and a simple multilayer perceptron (MLP). For the Transformer, each state-action pair in a segment is encoded and passed through an MLP head to produce per-step advantage values $A_\phi(s_k, a_k), \ldots, A_\phi(s_{k+N-1}, a_{k+N-1})$, which are then aggregated via summation to obtain $A_\phi(\tau)$. The MLP directly maps individual state-action pairs to scalar advantage values. To prevent overfitting, we employ early stopping based on validation accuracy or terminate training once a predefined maximum number of update steps is reached. Section 4 compares the performance of both architectures.

## 3.2. Policy Update

We use the advantage function $A_\phi(\tau)$ to first extract an optimal segment distribution $p^*(\tau)$ such that segments with higher advantage have higher likelihood while staying close to the data distribution. Subsequently, we infer the policy $\pi(a|s)$ from $p^*(\tau)$ using a maximum likelihood fitting step.

**Obtaining the segment distribution.** We optimize the segment distribution using a constrained optimization problem in the segment space $\tau$,

$$\max_p \int p(\tau) A_\phi(\tau) d\tau$$
$$\text{s.t. } \text{KL}\left(p(\tau) \,\|\, p_D(\tau)\right) \le \epsilon \text{ and } \int p(\tau) d\tau = 1. \quad (3)$$

This optimization problem has two properties that are crucial in our setting. First, it includes a trust-region that constrains the new segment distribution to be close to the data distribution $p_D$, and second, finding a new segment distribution is a pure offline process and does not involve generating new segments using the policy. Both benefits are important in the offline policy optimization step to avoid out-of-distribution samples (Kostrikov et al., 2021; Nair et al., 2020). Here, we leverage this formalism to extract a reweighted segment distribution using preference data only. The derivations in this section adapt the relative entropy policy search framework of Peters et al. (2010) to segment-wise updates in the offline preference setting. Using Lagrangian optimization (Boyd & Vandenberghe, 2004), we can obtain the optimal solution to the optimization problem in Equation (3) that we summarize in the following proposition.

**Proposition 3.1** (Optimal Segment Distribution). *(Peters et al., 2010) Given an advantage function $A_\phi(\tau) = \sum_t A_\phi(s_t, a_t)$, samples from a preference distribution $\tau \sim p_D(\tau)$ and a fixed Lagrange multiplier $\lambda$, the optimal policy to the optimization problem in Equation (3) is*

$$p^*(\tau) \propto p_D(\tau) \exp\left(\frac{1}{\lambda}A_\phi(\tau)\right). \quad (4)$$

Proposition 3.1 shows that the new segment distribution $p^*$ is proportional to the preference distribution $p_D$, but reweighted with the exponential of the advantage estimate of the segment. Intuitively, it increases the likelihood of the data distribution where the advantage is high. In Section A.1, we provide the detailed derivation for the segment-based case as considered here.

**Extracting the policy from the segment distribution.** We can obtain a policy $\pi_\theta$ by fitting the segment distribution

$$p_\theta(\tau) = p_D(s_0) \prod_{t=0}^{N-1} p(s_{t+1}|s_t, a_t)\pi_\theta(a_t|s_t) \quad (5)$$

induced by policy $\pi_\theta$ to the desired segment distribution $p^*(\tau)$. This can be achieved by minimizing the KL-projection

$$\theta^* = \text{argmin}_\theta \quad \text{KL}(p^*(\tau) \,\|\, p_\theta(\tau)).$$

After simplifying and re-arranging terms (see Section A.3), we can realize that minimizing the projection is equivalent to maximizing the following objective

$$\mathcal{L}(\theta) = \mathbb{E}_{p_D}\left[w(\tau) \log p_\theta(\tau)\right], \quad (6)$$

where the importance weights

$$w(\tau) = \frac{p^*(\tau)}{p_D(\tau)} \propto \exp\left(\frac{1}{\lambda}A_\phi(\tau)\right) \quad (7)$$

are given by the exponentially transformed segment advantages. We can further simplify this optimization by using Equation (5) to factorize $p_\theta(\tau)$ and, as the log turns the product into sums, removing all terms that do not depend on $\theta$. Hence, we arrive at the following weighted maximum likelihood problem

$$\mathcal{L}(\theta) = \sum_{\tau \in D} \sum_{(s_t, a_t) \in \tau} \exp\left(\frac{A_\phi(\tau)}{\lambda}\right) \log \pi_\theta(a_t|s_t). \quad (8)$$

Similar optimization schemes for obtaining a parametric policy were suggested in online and offline RL algorithms, for example, by Nair et al. (2020); Abdolmaleki et al. (2018). However, these schemes typically use weightings that are computed from single state-action pairs instead of advantages of whole segments.

### 3.3. Computing the Lagrange Multiplier $\lambda$

Manually setting the Lagrange multiplier $\lambda$ can be difficult, because for too high $\lambda$ values, the importance weights in Equation (7) become uniform, forcing the optimization to consider all data points, including undesired bad examples. Too small $\lambda$ values exploit data points with high advantage, such that the effectively used number of data points collapses to only a few.

The Lagrange multiplier $\lambda$ directly depends on the parameter $\epsilon$ that upper bounds how much the policy $\pi$ is allowed to deviate from the preference distribution $p_D$. Finding a good value for $\lambda$ is therefore crucial to balance the exploitation of data points with high advantage values, especially in scenarios, where data is scarce and the advantage function $A_\phi(\tau)$ might not provide accurate estimates. We follow prior works (Peters et al., 2010; Daniel et al., 2016) and optimize for $\lambda$ by minimizing the dual function as stated in the next proposition.

**Proposition 3.2** (Optimal Lagrange multiplier). *(Peters et al., 2010) Minimizing the dual function*

$$g(\lambda) = \lambda\epsilon + \lambda \log \int p_D(\tau) \exp\left(\frac{A_\phi(\tau)}{\lambda}\right) d\tau \quad (9)$$

*yields the optimal Lagrange multiplier $\lambda^*$.*

Section A.2 provides a detailed derivation. The expectation over $p_D(\tau)$ in the dual function can be easily approximated via a Monte Carlo estimate using the available samples.

The dual function optimization directly connects the choice of $\lambda$ with the KL-bound $\epsilon$. However, tuning $\epsilon$ can often be problem-specific as the KL depends on the problem's action dimensionality. Moreover, good KL values typically depend on the dataset size, where larger datasets typically allow for larger step sizes and thus KL values than small datasets. Hence, instead of tuning the KL bound $\epsilon$ by hand,

we propose to automatically adapt $\epsilon$ based on the desired number of effective samples

$$n_{\text{eff}} = \frac{\left(\sum_i w_i\right)^2}{\sum_i w_i^2}, w_i = \exp\left(\frac{1}{\lambda}\sum_t A_\phi(s_t^i, a_t^i)\right), \quad (10)$$

which is measured based on the importance weights $w_i$ and therefore depends on $\lambda$. For a desired value $n_{\text{eff}}^*$ we can automatically find the corresponding $\epsilon$ through an iterative process, as shown in Algorithm 1. Choosing a value for $n_{\text{eff}}^*$ is often more intuitive than defining $\epsilon$, as the number of effective samples corresponds to the number of samples that are effectively used for updating the policy parameters $\theta$. For example, a value of $n_{\text{eff}}^* = 10\%$ for a dataset with $500$ preferences would mean that segments from $50$ preferences meaningfully contribute to each policy update. This interpretation is more straightforward than defining an upper bound on the trust region $\epsilon$ of the KL constraint, whose value depends on the dimensionality of the action space.

## 4. Evaluations

We analyze the learning behavior of PAWS using two different parameterizations for the advantage function, a Transformer-based and MLP-based advantage function, which we denote as PAWS (Trans.) and PAWS (MLP), respectively. As baselines, we evaluate Behavior Cloning (BC), P-IQL, CPL (Hejna et al., 2024), CPL+KL, which is related to PPL (Cho et al., 2025), Preference Transformer (Kim et al., 2023), and IPL (Hejna & Sadigh, 2023). IQL (Kostrikov et al., 2021) has been used to optimize a utility function in the PbRL setting in several recent works (Hejna et al., 2024; Kim et al., 2023) using slightly different objectives. In our case, we implement it using the advantage as the target (Hejna et al., 2024). Following previous naming conventions, we call the resulting baseline P-IQL. We additionally evaluate DPPO (An et al., 2023), which is designed to rely on a large unlabeled trajectory dataset alongside preferences. Without this auxiliary dataset, which is not available in our offline setup, DPPO underperforms all other baselines on every task we evaluated, achieving e.g. only $9\%$ success on Sweep Into and $1\%$ on Peg Insert Side with 500 preferences. We therefore relegate the full DPPO results to Section D and exclude DPPO from the main tables for clarity. We further compare performance under two preference budgets, namely $50$ and $500$ preferences, sampled from the same larger dataset. We follow the sparse sampling setup proposed in CPL (Hejna et al., 2024). From a pool of $K$ segments, we form each preference pair by drawing one segment from the first half and one from the second half. Our evaluations are performed on 10 different Meta-World tasks (Yu et al., 2020) and four locomotion tasks from D4RL (Fu et al., 2020; Towers et al., 2024). More details about these tasks are provided in Section E.

*Table 1.* **Meta-World:** Task success (%) $\pm$ 2SE. Best results for $n = 50$ and $n = 500$ are highlighted with orange and teal, respectively. The last two rows report the average performance across tasks and the relative improvement (%) over Behavior Cloning (BC).

| Task | BC | | P-IQL | | CPL | | CPL+KL | | Pref Trans. | | IPL | | PAWS (Trans.) | | PAWS (MLP) | |
|---|---|---|---|---|---|---|---|---|---|---|---|---|---|---|---|---|
| #Preferences | 50 | 500 | 50 | 500 | 50 | 500 | 50 | 500 | 50 | 500 | 50 | 500 | 50 | 500 | 50 | 500 |
| Button Press | 69±3 | 67±3 | 72±6 | 77±5 | 70±8 | 87±5 | 67±7 | 86±3 | 71±5 | 77±2 | 66±8 | 69±4 | 80±5 | 84±4 | 82±6 | 82±4 |
| Door Open | 48±6 | 52±3 | 52±8 | 83±3 | 36±9 | 71±9 | 44±11 | 77±6 | 62±6 | 87±2 | 47±8 | 59±4 | 70±8 | 96±1 | 65±15 | 98±1 |
| Drawer Open | 54±5 | 60±6 | 58±6 | 71±2 | 53±5 | 79±3 | 56±7 | 78±5 | 45±3 | 71±1 | 44±5 | 55±3 | 44±11 | 74±3 | 40±12 | 75±3 |
| Faucet Close | 51±6 | 66±3 | 59±7 | 80±3 | 53±7 | 64±4 | 48±6 | 63±3 | 59±3 | 85±2 | 54±6 | 67±4 | 67±10 | 87±3 | 68±8 | 87±3 |
| Lever Pull | 36±7 | 44±3 | 31±5 | 38±4 | 28±8 | 46±3 | 29±8 | 47±4 | 34±3 | 47±2 | 35±4 | 54±3 | 28±5 | 58±5 | 30±12 | 55±4 |
| Peg Insert Side | 33±7 | 48±3 | 31±6 | 78±5 | 32±6 | 68±4 | 27±8 | 67±4 | 33±3 | 80±1 | 29±5 | 49±3 | 24±7 | 81±4 | 23±9 | 82±3 |
| Plate Slide | 47±5 | 50±6 | 50±4 | 74±7 | 42±5 | 65±3 | 42±7 | 65±5 | 55±3 | 78±2 | 54±8 | 56±3 | 48±6 | 74±5 | 49±9 | 78±5 |
| Push Back | 25±4 | 37±2 | 23±5 | 43±3 | 25±7 | 45±2 | 18±5 | 43±4 | 25±2 | 48±2 | 26±4 | 38±2 | 26±5 | 56±3 | 24±4 | 53±3 |
| Sweep Into | 30±6 | 58±3 | 35±5 | 66±5 | 26±9 | 66±5 | 31±6 | 64±6 | 31±2 | 66±2 | 31±5 | 57±2 | 35±8 | 74±3 | 36±9 | 74±4 |
| Window Close | 69±8 | 91±2 | 78±9 | 97±1 | 64±7 | 85±5 | 59±8 | 83±4 | 86±0 | 96±1 | 71±6 | 94±2 | 94±5 | 98±1 | 91±5 | 99±0 |
| Avg. over tasks | 46.2 | 57.3 | 48.9 | 70.7 | 42.9 | 67.6 | 42.1 | 67.3 | 50.1 | 73.5 | 45.7 | 59.8 | 51.6 | 78.2 | 50.8 | 78.3 |
| Improvement (%) | 0.0 | 0.0 | 5.8 | 23.4 | -7.1 | 18.0 | -8.9 | 17.5 | 8.4 | 28.3 | -1.1 | 4.4 | 11.7 | 36.5 | 10.0 | 36.6 |

For each task, we construct a dataset containing an equal number of samples collected from four policies of varying quality. This data generation procedure more realistically reflects differences in human expertise, as commonly observed in settings such as teleoperation or human-in-the-loop control. In contrast, several baseline methods (Hejna et al., 2024; Cho et al., 2025) generate suboptimal behavior by injecting uncorrelated Gaussian noise into expert actions, an assumption that is unlikely to capture structured deviations arising from lower skill levels. To obtain policies of different quality, we train an expert policy using Soft Actor-Critic (SAC) (Haarnoja et al., 2018) and periodically save checkpoints throughout training. From these checkpoints, we automatically select four policies whose performance spans a broad range, from poor to high-quality behavior. A detailed performance evaluation of the resulting policies is provided in Table 11 in Section E.

Preference labels are generated using the highest-performing policy as a proxy for an expert annotator. We derive preferences from the policy's log action probabilities. Under SAC, the optimal policy follows a Boltzmann distribution with respect to the advantage function, such that log probabilities correspond to advantage values (Haarnoja et al., 2017). This induces a ranking based on relative action quality rather than absolute Q-values, which better reflects how humans compare behaviors. Consequently, log probabilities provide a principled and efficient surrogate for expert preference labels.

**Meta-World Tasks.** We modified the Meta-World tasks in two ways, as proposed in our baseline (Hejna et al., 2024).

Namely, we randomize the initial hand position and also remove proprioceptive history from the observation. The success rates on Meta-World tasks are reported in Table 1, and the mean returns are provided in Table 6 in Section B. Our evaluation methodology follows CPL. Specifically, we execute 25 rollouts per evaluation point, smooth the results by averaging over eight adjacent points, and report the peak performance. All experiments are conducted over 10 random seeds, and we report the mean along with two times the standard error. All methods use the same MLP policy architecture; for PAWS, we fix the effective sample size to $n_{\text{eff}} = 10\%$ across all evaluations. For the baselines, we use the implementations and recommended hyperparameters provided by Hejna et al. (2024) (see Section F). Overall, our method achieves higher average success rates and higher mean returns across tasks (Table 6). In the low-data regime with only 50 preference queries, several baseline methods degrade in performance and, in some cases, perform worse than behavior cloning. In contrast, PAWS outperforms behavior cloning and all other baselines in both low- and high-data settings in the majority of the tasks.

**Locomotion Tasks.** We evaluate four different locomotion tasks, namely Ant, HalfCheetah, Hopper, and Walker2d. The average returns from 25 rollouts over 10 seeds are presented in Table 2. The evaluation procedure is the same as for the Meta-World tasks. Moreover, we use the same hyperparameters and policy network size, and we use $n_{\text{eff}} = 30\%$ for PAWS for all evaluations. On all locomotion tasks, PAWS achieves higher returns than all of the baselines for both preference budgets.

*Table 2.* **Locomotion Tasks:** Average episode returns over 25 rollouts and 10 random seeds $\pm$ 2SE. Best results for $n{=}50$ and $n{=}500$ preferences are highlighted with orange and teal, respectively. Both PAWS variants outperform all baselines across all four locomotion tasks and both preference budgets. The same hyperparameters and policy architecture are used for all methods, with $n_{\text{eff}}^{*} = 30\%$ for PAWS.

| Task | BC | | P-IQL | | CPL | | CPL+KL | | Pref Trans. | | IPL | | PAWS (Trans.) | | PAWS (MLP) | |
|---|---|---|---|---|---|---|---|---|---|---|---|---|---|---|---|---|
| #Preferences | *50* | *500* | *50* | *500* | *50* | *500* | *50* | *500* | *50* | *500* | *50* | *500* | *50* | *500* | *50* | *500* |
| Ant | 546±43 | 593±34 | 567±37 | 847±12 | 508±34 | 563±33 | 518±22 | 581±8 | 505±17 | 784±17 | 384±16 | 634±10 | 685±22 | 850±10 | 707±26 | 882±16 |
| HalfCheetah | 1019±29 | 1085±66 | 1029±20 | 1031±107 | 998±45 | 950±87 | 998±35 | 948±93 | 945±25 | 968±85 | 690±3 | 1118±26 | 1063±26 | 1483±92 | 810±93 | 1450±251 |
| Hopper | 456±45 | 401±22 | 397±39 | 571±65 | 391±39 | 398±35 | 451±61 | 405±32 | 500±28 | 552±48 | 436±85 | 385±48 | 484±41 | 563±60 | 512±57 | 637±50 |
| Walker2d | 167±33 | 266±30 | 116±33 | 947±32 | 119±28 | 446±55 | 123±31 | 476±77 | 150±40 | 726±56 | 160±76 | 251±34 | 208±82 | 997±21 | 235±70 | 1023±16 |

**Human-Labeled Data.** Humans may provide preferences that differ from those of an oracle (Lee et al., 2021a). Therefore, for two Meta-World tasks, Button Press and Door Open, we collected 50 pairwise comparisons per task from each of 10 non-author human labelers. Further details and statistical analysis, as well as the Graphical User Interface used to collect the preferences, are provided in Section H.

PAWS and all the baselines used the same hyperparameters as in the main evaluation (Table 1). Each of the 10 seeds was trained on the 50 comparisons provided by a single distinct labeler, so the reported variance reflects both seed and labeler variability. The mean success rates over 10 seeds are presented in Table 3. Overall, the results are in line with the preferences provided by the oracle and show the benefits of using PAWS, with PAWS (MLP) achieving the highest success rate on both tasks and PAWS (Trans.) ranking second on Door Open.

**Varying the Number of Effective Samples.** As discussed in Section 3.3, the effective sample size $n_{\text{eff}}$ controls how far the updated policy is allowed to deviate from the provided data distribution. Larger values of $n_{\text{eff}}$ enforce higher overlap between the updated policy and the segment distribution, while smaller values permit more selective updates that emphasize only segments with high advantage.

We study the effect of $n_{\text{eff}}$ across different data regimes. In Figure 3a, we consider a higher-data setting with 500 preferences and report aggregated success rates over the *Peg Insert Side*, *Sweep Into*, and *Drawer Open* tasks for six relative values of $n_{\text{eff}}$, ranging from $5\%$ to $50\%$ of the available samples. In this regime, performance improves as the effective sample size decreases, peaking at $n_{\text{eff}} = 10\%$, indicating that with sufficient data coverage the policy can benefit from more aggressive, advantage-driven updates.

In contrast, Figure 3b shows results for the same range of relative effective sample sizes on the same three tasks using 50 preferences. In this low-data regime, performance is maximized at a substantially larger value of $n_{\text{eff}}$. In this case, using too few effective samples leads to poor estimation of the policy, whereas larger values of $n_{\text{eff}}$ keep the policy closer to the provided data distribution and yield more stable improvements.

Taken together, these results suggest that larger datasets permit more aggressive policy updates, whereas smaller datasets benefit from conservative updates that rely on broader data support. The observed trend further indicates that the absolute number of effective samples, rather than their relative proportion, may provide a more robust criterion for controlling policy updates across different data regimes. We consider this a promising direction for future work. In this study, we fix the effective sample size as a relative proportion of the available data to simplify evaluation and ensure consistent behavior across tasks with different dataset sizes. Despite this simplification, the chosen hyperparameters yield strong and stable performance across all experiments, although further gains for PAWS may be achievable with adaptive or absolute effective sample size selection, as suggested by the analysis in Figure 3a and Figure 3b. We note that results in these figures are averaged over 5 random seeds and may therefore differ slightly from those reported in Table 1.

**Segment- and State-action-based Updates.** In addition to the results reported in Table 1, we explicitly compare segment-level policy updates with state-action-level updates that rely on the learned advantage function. This comparison directly evaluates the effect of the training and inference distribution shift discussed earlier: while the advantage function is trained on segment-level preference data, state-action-based updates require querying it on individual time steps, thereby reintroducing the mismatch present in standard PbRL methods. We further consider two architectures for the advantage function, parameterized either by an MLP or a Transformer. Aggregated results across all tasks are reported in Table 4. Consistent with our analysis, PAWS achieves substantially higher performance than policies updated using single-step advantages. Additional per-task results are provided in Table 7 and Table 8 in Section C.

We note that our setting assumes action quality is correlated over time, reflected by our use of different policies with varying performance for data collection. We argue this is more realistic than datasets that simulate varying data quality by adding Gaussian noise to individual actions (Hejna

*Table 3.* Task success (%) $\pm$ 2SE with **human-collected preferences** from 10 non-author participants on Button Press and Door Open tasks. Each participant labeled 50 pairwise comparisons per task.

| Task | BC | P-IQL | CPL | CPL+KL | Pref Trans. | IPL | PAWS (Trans.) | PAWS (MLP) |
|------|------|-------|------|--------|-------------|-----|---------------|------------|
| Button Press | 44.7±2.4 | 53.2±3.7 | 47.5±7.4 | 49.0±10.1 | 55.2±5.6 | 47.8±4.2 | 57.2±5.6 | 56.2±5.6 |
| Door Open | 73.6±2.4 | 64.0±3.7 | 55.6±9.2 | 54.8±8.0 | 76.9±4.4 | 76.2±3.3 | 82.2±3.8 | 86.0±3.0 |

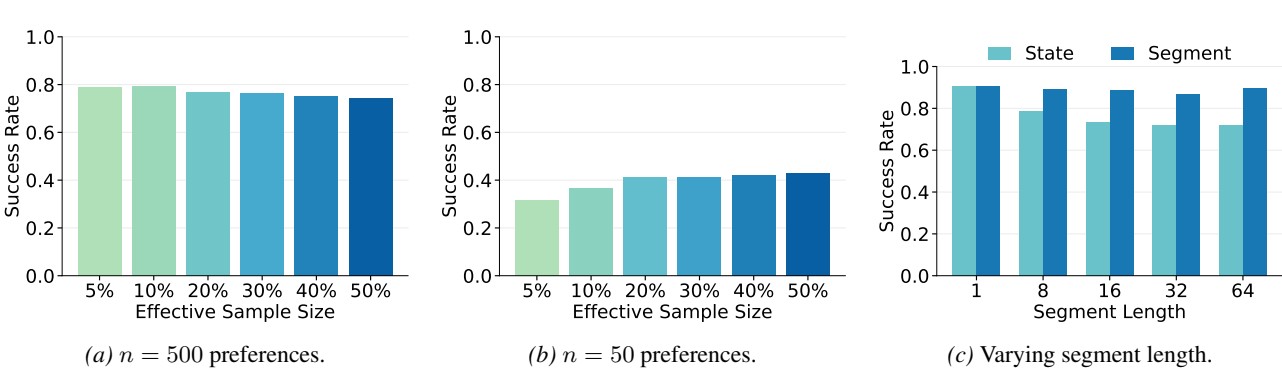

*(a)* $n = 500$ preferences.    *(b)* $n = 50$ preferences.    *(c)* Varying segment length.

*Figure 3.* **Ablations on the (a)-(b) Number of Effective Samples and the (c) Segment Length**. **(a)-(b)** Effect of Number of Effective Samples for **(a)** 500 Preferences and for **(b)** 50 Preferences aggregated over the *Peg Insert Side*, *Sweep Into*, and *Drawer Open* tasks, each with 5 seeds. The results suggest that for a high number of available data points **(a)**, a smaller number of effective samples leads to improved results, allowing the policy to deviate more from the data distribution's support. For smaller number of preferences **(b)**, staying close to the data distribution with a higher relative number of effective samples is beneficial. **(c)** Increasing segment length amplifies the *distribution shift* when using single-timestep advantages for policy optimization, leading to worse performance. This problem is absent when using segment-based advantages for the policy update. Experiments are done on 5 seeds, and the same number of state-action pairs is used for all segment lengths.

*Table 4.* Aggregated performance (success rates %) across all 10 Meta-World tasks for two architectures (MLP, Transformer) and two policy update granularities (segment-level, used by PAWS, and state-action level, denoted **State**).

| #Pref. | PAWS(MLP) | PAWS(Trans.) | State(MLP) | State(Trans.) |
|--------|-----------|--------------|------------|---------------|
| *50* | 51 | **52** | 40 | 44 |
| *500* | **78** | **78** | 67 | 63 |

et al., 2024). Under uncorrelated noise, within-segment quality would be highly variable and single-action importance weights would likely perform better. In the multi-expert scenario considered here, however, segment-level weights clearly outperform single-step weights.

**Varying the Segment Length.** We evaluate our method across different segment lengths and compare segment-wise policy updates with state-action-based updates. When the segment length is one, both settings coincide. In practice, however, such short segments are insufficient for reliable preference assessment, as they provide limited contextual information about behavior quality. For all experiments, we fix the total number of segments to 1,000 and generate preference datasets by uniformly sampling segment pairs. To ensure a fair comparison across segment lengths and maintain an approximately equal number of state-action samples for training the advantage function, we increase the number of sampled preferences as the segment length decreases. For

instance, for segments of length 64, we sample 10,000 preference pairs, while for segments of length 32, we sample 20,000 preference pairs. Results averaged over 10 Meta-World tasks and 5 random seeds are reported in Figure 3c. As can be seen, state-action-based updates degrade sharply once the segment length exceeds one, with the largest drop occurring up to a length of roughly 16 and performance remaining low for longer segments, reflecting the training and inference distribution shift. In contrast, the performance of our method remains substantially more stable, demonstrating that segment-wise policy updates effectively mitigate this degradation and highlighting the benefits of preserving segment-level preference information.

To further isolate the role of segment length at the policy-update stage, we additionally train a Transformer-based advantage function on 500 preferences of segment length 64 and then perform policy updates with progressively shorter segments. As shown in Table 5, the success rate decreases monotonically as the update-time segment length shrinks. This confirms that shorter segments at update time degrade performance even when the advantage function itself is trained on longer, more informative segments, reinforcing the importance of operating at the segment level throughout both training and policy optimization.

**Spearman's Rank Correlation for Segment- and State-action-based Updates.** To assess temporal credit assign-

*Table 5.* The Transformer-based advantage function is trained once on 500 preferences, then used to update the policy with progressively shorter segments. Averaged over 10 Meta-World tasks and 5 seeds. The 64 column corresponds to the setting reported in Table 1.

| Segment length | 64 (Table 1) | 32 | 16 | 8 |
|---|---|---|---|---|
| **Success Rate [%]** | 78.2 | 75.8 | 69.9 | 64.3 |

ment, we measure how well learned policies preserve the expert's relative ranking of trajectory segments using Spearman's rank correlation coefficient $r_s$. This metric computes the Pearson coefficient for the rankings of two vectors. More details are presented in Section G. In our case, for each task, we compute the action likelihoods at each state under three policies: the expert SAC policy, the policy trained using our segment-based updates, and the policy trained using single-step updates.

For each trajectory segment, we aggregate the corresponding likelihoods and compute Spearman's rank correlation between the expert policy and each learned policy. This yields two correlation values per segment, one for the segment-based update and one for the state-action-based update. Higher correlation indicates better preservation of the expert's preference ordering and, consequently, more effective temporal credit assignment. We evaluate all tasks using 5 random seeds and 500 preference queries. Overall, the segment-based update used in PAWS achieves a higher mean Spearman's rank correlation coefficient ($r_s^{seg.} = 0.22$) than the step-based approach ($r_s^{step} = 0.05$), indicating that segment-level updates better preserve the expert's ranking over behaviors. Per-task results are reported in Section G.

## 5. Conclusion

We propose PAWS, a novel method for preference learning that first learns an advantage function and subsequently updates the policy by exploiting it. Notably, PAWS updates the policy based on segment data rather than single state-action pairs to mitigate the *temporal credit assignment problem* that arises because the advantage function is learned on whole segments rather than single state-action pairs. Overall, the proposed method achieves superior performance on various simulated manipulation and locomotion tasks, with preferences coming from both a simulated oracle and real humans. Our results further indicate that the gains stem primarily from aligning the training and inference distributions of the learned utility, rather than from the specific architectural choice of the advantage function. Both the MLP and Transformer variants substantially outperform their per-step counterparts, and the benefit is particularly pronounced in the low-data regime, where step-level utility estimates are noisiest.

**Limitations.** PAWS assigns the same importance weight to every state-action pair within a segment (Equation (8)), which is well suited to the realistic setting of temporally correlated behavior quality but less effective when a segment mixes high- and low-quality actions, where finer-grained per-step weighting would be preferable. A second limitation is the need to choose the effective sample size $n_{\text{eff}}^*$. While it is more intuitive than directly tuning the KL bound $\epsilon$, our ablation suggests that the optimal *relative* value depends on dataset size, with smaller datasets favoring more conservative updates and larger datasets benefiting from more aggressive updates that selectively focus on high-advantage segments. Finally, our evaluation is restricted to simulated robotic tasks with oracle and small-scale human preference data. Extending the analysis to large-scale, noisier human feedback remains an important next step.

**Future Work.** A natural direction is to relax the uniform per-segment weighting by introducing a learned per-step modulation that respects the segment-level training distribution, for example through attention-based reweighting within a segment. We plan to extend PAWS to the offline-to-online case, where newly collected preference data continuously refines both the advantage function and the policy update, enabling preference-based fine-tuning of pretrained models. A third direction is to replace the fixed relative $n_{\text{eff}}^*$ with an absolute, data-adaptive criterion, as suggested by the trend observed in Figure 3a and Figure 3b. A further direction is to study PAWS under realistic preference noise, including non-stationary annotator behavior. Combining segment-level updates with active query selection could focus limited annotator effort on the segments whose advantage estimates would benefit most, improving sample efficiency.

Beyond robotics, the segment-level update perspective directly maps to preference fine-tuning of large generative models, where comparisons are typically expressed over entire responses rather than individual tokens. Standard RLHF pipelines train a per-token reward model from response-level preferences and then optimize the policy with per-token PPO, recreating the same training and inference mismatch we identify here.

## Impact Statement

This paper proposes a novel preference-based reinforcement learning method evaluated on robotic manipulation and locomotion tasks in simulation. Such methods may reduce the need for manually designed reward functions, but can inherit biases present in preference signals provided by humans. All experiments are conducted in simulated environments, and we do not anticipate immediate negative societal impacts from this work.

## Acknowledgements

We thank the anonymous reviewers for their valuable feedback and suggestions. The work has been financially supported by the German Research Foundation (DFG, Deutsche Forschungsgemeinschaft) as part of the SFB-1574 – 471687386 "Circular Factory", and by the European Research Council (ERC) under the European Union's Horizon Europe programme through the project SMARTI³ (Grant Agreement No. 101171393). This work has been supported by the German Federal Ministry of Research, Technology, and Space (BMFTR) under the Robotics Institute Germany (RIG). The authors acknowledge support from the InnovationCampus Future Mobility ICM, and the state of Baden-Württemberg through bwHPC, as well as the HoreKa supercomputer, funded by the Ministry of Science, Research and the Arts Baden-Württemberg and the German Federal Ministry of Education and Research.

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

# A. Proofs

## A.1. Proof of the Optimal Distribution

*Proof.* The Lagrangian to the optimization problem in Equation (3) is given by

$$
\begin{aligned}
L(p, \lambda, \beta) &= \int p(\tau) A_\phi(\tau) \, d\tau + \lambda \left( \epsilon - \int p(\tau) \big( \log p(\tau) - \log p_D(\tau) \big) d\tau \right) \\
&\quad + \beta \left( 1 - \int p(\tau) \, d\tau \right) \\
&= \int p(\tau) \Big( A_\phi(\tau) - \lambda \big( \log p(\tau) - \log p_D(\tau) \big) - \beta \Big) d\tau + \lambda \epsilon + \beta.
\end{aligned}
\tag{11}
$$

Obtaining the optimal solution for fixed Lagrange multipliers $\lambda, \beta$ yields

$$
\begin{aligned}
0 &\overset{!}{=} \nabla_p L = A_\phi(\tau) - \lambda \log p(\tau) + \lambda \log p_D(\tau) - \lambda - \beta \\
&\to p^* = \exp \left( \frac{A_\phi(\tau) + \lambda \log p_D(\tau)}{\lambda} \right) \exp \left( \frac{-\beta - \lambda}{\lambda} \right)
\end{aligned}
\tag{12}
$$

$\square$

## A.2. Proof of the Dual Function

*Proof.* We can eliminate the Lagrange multiplier $\beta$ to the normalization constraint

$$
\int p(\tau) \, d\tau = 1
\tag{13}
$$

in the optimization problem Equation (3) by inserting the optimal solution in Equation (12) into the Equation (13), such that we obtain

$$
\begin{aligned}
1 &= \exp \left( \frac{-\beta - \lambda}{\lambda} \right) \int \exp \left( \frac{A_\phi(\tau) + \lambda \log p_D(\tau)}{\lambda} \right) d\tau \\
0 &= \frac{-\beta - \lambda}{\lambda} + \log \int \exp \left( \frac{A_\phi(\tau) + \lambda \log p_D(\tau)}{\lambda} \right) d\tau \\
\beta &= -\lambda + \lambda \log \int \exp \left( \frac{A_\phi(\tau) + \lambda \log p_D(\tau)}{\lambda} \right) d\tau
\end{aligned}
\tag{14}
$$

By inserting the optimal solution $p^*$ in Equation (12) and $\beta$ in Equation (14) into the Lagrangian in Equation (11), we obtain

$$
\begin{aligned}
g(\lambda) &= \int p^*(\tau) \Big( A_\phi(\tau) - A_\phi(\tau) - \lambda \log p_D(\tau) + \beta + \lambda + \lambda \log p_D(\tau) - \beta \Big) d\tau + \lambda \epsilon + \beta \\
&= \int p^*(\tau) \lambda \, d\tau + \lambda \epsilon + \beta \\
&= \lambda + \lambda \epsilon + \beta \\
&= \lambda \epsilon + \lambda \log \int \exp \left( \frac{A_\phi(\tau) + \lambda \log p_D(\tau)}{\lambda} \right) d\tau \\
&= \lambda \epsilon + \lambda \log \int p_D(\tau) \exp \left( \frac{A_\phi(\tau)}{\lambda} \right) d\tau.
\end{aligned}
\tag{15}
$$

$\square$

## A.3. Proof of Policy Extraction

*Proof.* The moment-projection of the non-parametric optimal distribution $p^*$ onto the set of parametrizable distributions,

$$\theta^* = \text{argmin}_\theta \, D_{\text{KL}}\big(p^*(\tau) \,\|\, p_\theta(\tau)\big),$$

is a likelihood maximization as the KL objective,

$$
\begin{aligned}
D_{\text{KL}}\big(p^*(\tau) \,\|\, p_\theta(\tau)\big) &= \int p^*(\tau) \log \frac{p^*(\tau)}{p_\theta(\tau)} d\tau = \int p_D(\tau) \frac{p^*(\tau)}{p_D(\tau)} \log \frac{p^*(\tau)}{p_\theta(\tau)} d\tau \\
&= \int p_D(\tau) w(\tau) \log p^*(\tau) \, d\tau - \int p_D(\tau) w(\tau) \log p_\theta(\tau) \, d\tau \\
&= C(p_D, p^*) - \mathbb{E}_{p_D}\big[w(\tau) \log p_\theta(\tau)\big],
\end{aligned}
$$

equals the importance sampled maximum likelihood objective, $\mathcal{L}(\theta) = \mathbb{E}_{p_D}\big[w(\tau) \log p_\theta(\tau)\big]$, up to negation and a parameter-independent constant $C(p_D, p^*)$. Thus we have

$$\theta^* = \text{argmin}_\theta \, D_{\text{KL}}\big(p^*(\tau) \,\|\, p_\theta(\tau)\big) = \text{argmax}_\theta \, \mathbb{E}_{p_D}\big[w(\tau) \log p_\theta(\tau)\big] = \text{argmax}_\theta \, \mathcal{L}(\theta).$$

The importance sampling weights $w(\tau)$ directly follow from Equation (4), yielding

$$w(\tau) = \frac{p^*(\tau)}{p_D(\tau)} \propto \exp\left(\frac{1}{\lambda} A_\phi(\tau)\right),$$

or more specifically using Equation (12),

$$
\begin{aligned}
w(\tau) &= \frac{\exp\left(\frac{A_\phi(\tau)}{\lambda}\right) \exp\left(\log p_D(\tau) - \frac{\beta + \lambda}{\lambda}\right)}{p_D(\tau)} \\
&= \exp\left(\frac{A_\phi(\tau)}{\lambda}\right) \exp\left(\frac{-\beta - \lambda}{\lambda}\right) =: \exp\left(\frac{A_\phi(\tau)}{\lambda}\right) \cdot Z,
\end{aligned}
\tag{16}
$$

for the normalization constant $Z$. Inserting Equation (5) into Equation (6),

$$
\begin{aligned}
\mathbb{E}_{p_D}\big[w(\tau) \log p_\theta(\tau)\big] &= \mathbb{E}_{p_D}\left[w(\tau) \log\left(p_D(s_0) \prod_{t=0}^{N-1} p(s_{t+1}|s_t, a_t) \pi_\theta(a_t|s_t)\right)\right] \\
&= \mathbb{E}_{p_D}\left[w(\tau) \left(\log p_D(s_0) + \sum_{t=0}^{N-1} \big(\log p(s_{t+1}|s_t, a_t) + \log \pi_\theta(a_t|s_t)\big)\right)\right] \\
&= \mathbb{E}_{p_D}\left[w(\tau) \sum_{t=0}^{N-1} \log \pi_\theta(a_t|s_t) + C(\tau)\right],
\end{aligned}
$$

considering that $C(\tau)$ is independent of the optimized parameters $\theta$, and that the importance weight normalization constant $Z$ of Equation (16) is constant, the objective reduces to

$$\text{argmax}_\theta \, \mathbb{E}_{p_D}\big[w(\tau) \log p_\theta(\tau)\big] = \text{argmax}_\theta \, \mathbb{E}_{p_D}\left[\sum_{t=0}^{N-1} \exp\left(\frac{A_\phi(\tau)}{\lambda}\right) \log \pi_\theta(a_t|s_t)\right].$$

$\square$

## A.4. Derivation of the Gradient

*Proof.* We can rewrite the more common state-action policy optimization counterpart to the maximum likelihood objective from Equation (6) as

$$\mathcal{L}_{ML}(\theta) = \mathbb{E}_{p_D}\left[\frac{1}{Z}\exp\left(\frac{1}{\lambda}\sum_t A_\phi(s_t, a_t)\right)\left(\log p(s_0) + \sum_t \big(\log \pi_\theta(a_t|s_t) + \log p(s_{t+1}|s_t, a_t)\big)\right)\right], \quad (17)$$

which includes the unknown initial state distribution $p(s_0)$ and the transition dynamics $p(s_{t+1}|s_t, a_t)$ as well as the intractable normalization constant $Z = \int \exp\left(\frac{1}{\lambda}\sum_t A_\phi(s_t, a_t)\right) d\tau$. However, note that neither of these entities affects the optimization, as the $\theta$-independent log-terms vanish when calculating the gradient and the constant factor $1/Z$ does not change the optimum, yielding

$$\nabla_\theta \mathcal{L} = \mathbb{E}_{p_D}\left[\exp\left(\frac{1}{\lambda}\sum_t A_\phi(s_t, a_t)\right)\left(\sum_t \nabla_\theta \log \pi_\theta(a_t|s_t)\right)\right], \quad (18)$$

which is closely related to Kober & Peters (2008). □

# B. Average Returns

In Table 6, we present the results of mean episode returns corresponding to the experiment setup of Table 1. The relative performance ordering per task remains largely unchanged when compared to the ranking based on success rate in Table 1.

*Table 6.* Mean episode returns across methods and preference counts. Best results for $n = 50$ and $n = 500$ are highlighted with **orange** and **teal**, respectively.

| Task | BC | | P-IQL | | CPL | | CPL+KL | | Pref Trans. | | IPL | | PAWS (Trans.) | | PAWS (MLP) | |
|---|---|---|---|---|---|---|---|---|---|---|---|---|---|---|---|---|
| | 50 | 500 | 50 | 500 | 50 | 500 | 50 | 500 | 50 | 500 | 50 | 500 | 50 | 500 | 50 | 500 |
| Button Press | 1423±37 | 1593±40 | 1423±39 | 1590±53 | 1443±86 | 1598±55 | 1416±74 | 1581±29 | 1483±97 | 1626±30 | 1339±82 | 1584±33 | 1356±74 | 1568±25 | 1583±81 | 1722±22 |
| Door Open | 870±90 | 970±38 | 870±95 | 1593±43 | 970±126 | 1598±55 | 762±125 | 1581±29 | 1136±151 | 1527±57 | 890±98 | 1047±51 | 822±127 | 1568±25 | 1294±158 | 1819±27 |
| Drawer Open | 1508±59 | 1562±22 | 1508±80 | 1562±22 | 1463±78 | 1626±44 | 1472±78 | 1646±41 | 1255±82 | 1631±22 | 1259±53 | 1510±31 | 1481±61 | 1617±35 | 1104±194 | 1634±47 |
| Faucet Close | 1516±59 | 1692±31 | 1516±59 | 1692±31 | 1599±79 | 1900±27 | 1516±86 | 1662±61 | 1581±69 | 1935±46 | 1499±78 | 1713±49 | 1449±68 | 1661±64 | 1696±133 | 1997±37 |
| Lever Pull | 366±43 | 426±18 | 366±45 | 426±18 | 343±33 | 406±32 | 347±32 | 450±51 | 331±49 | 426±42 | 343±24 | 439±20 | 338±33 | 467±45 | 350±57 | 462±39 |
| Peg Insert Side | 882±87 | 1281±28 | 882±87 | 1281±28 | 670±107 | 1511±49 | 836±98 | 1426±74 | 708±109 | 1508±20 | 731±119 | 1230±32 | 814±125 | 1463±65 | 505±136 | 1545±47 |
| Plate Slide | 1049±85 | 1156±94 | 1049±89 | 1156±97 | 1095±78 | 1582±78 | 950±124 | 1395±108 | 1150±120 | 1617±62 | 1140±125 | 1218±92 | 955±104 | 1344±91 | 1070±130 | 1548±91 |
| Push Back | 312±56 | 484±39 | 312±56 | 484±39 | 270±65 | 607±111 | 261±88 | 629±45 | 262±53 | 698±62 | 300±58 | 490±47 | 203±68 | 584±55 | 363±73 | 872±70 |
| Sweep Into | 410±85 | 1001±54 | 410±90 | 1001±57 | 475±85 | 1117±103 | 325±123 | 1142±100 | 397±57 | 1109±59 | 410±77 | 900±54 | 439±111 | 1128±88 | 463±99 | 1326±85 |
| Window Close | 1095±102 | 1371±22 | 1095±97 | 1371±34 | 1198±104 | 1530±34 | 1015±106 | 1371±22 | 1107±229 | 1492±46 | 1034±109 | 1390±36 | 982±123 | 1358±47 | 1373±90 | 1572±17 |

## C. Comparison of Segment- and State-Based Updates

The detailed per-task success rates for the policy update type and architecture ablation can be found in Table 7 and Table 8. PAWS consistently outperforms the corresponding state-wise update for the same architecture with only few exceptions. In the $n = 50$ preferences case and applied to the Transformer network, a slightly lower success rate for *Lever Pull* and a tie for *Sweep Into* is achieved. When more preferences, $n = 500$, are available, only slight variations in success rate for *Lever Pull* break the trend.

*Table 7.* Performance results for individual tasks with **50** preferences. Values represent success rates (%) $\pm$ 2SE. Best results are highlighted with **bold**.

| Task | PAWS (MLP) | PAWS (Transformer) | State (MLP) | State (Transformer) |
| --- | --- | --- | --- | --- |
| Button Press | **82±6** | 80±5 | 72±6 | 74±5 |
| Door Open | 65±14 | **70±8** | 40±12 | 42±9 |
| Drawer Open | 39±12 | **44±11** | 22±10 | 31±13 |
| Faucet Close | **68±8** | 67±10 | 64±9 | 60±8 |
| Lever Pull | 30±12 | 28±6 | 16±11 | **32±9** |
| Peg Insert Side | 23±9 | **24±7** | 19±9 | 20±9 |
| Plate Slide | **49±9** | 48±6 | 37±6 | 45±8 |
| Push Back | 24±4 | **26±5** | 17±3 | 19±5 |
| Sweep Into | **36±9** | 34±8 | 33±6 | 34±5 |
| Window Close | 91±7 | **94±5** | 79±8 | 80±8 |

*Table 8.* Performance results for individual tasks with **500** preferences. Values represent success rates (%) $\pm$ 2SE. Best results are highlighted with **bold**.

| Task | PAWS (MLP) | PAWS (Transformer) | State (MLP) | State (Transformer) |
| --- | --- | --- | --- | --- |
| Button Press | 82±4 | **84±4** | 75±4 | 78±5 |
| Door Open | **98±1** | 96±1 | 82±4 | 64±8 |
| Drawer Open | **75±3** | 74±3 | 59±4 | 66±3 |
| Faucet Close | **87±3** | **87±3** | 71±10 | 73±5 |
| Lever Pull | 55±4 | **58±5** | 56±4 | 55±3 |
| Peg Insert Side | **82±3** | 81±4 | 68±5 | 55±4 |
| Plate Slide | **78±5** | 74±5 | 54±6 | 52±7 |
| Push Back | 53±3 | **56±3** | 44±4 | 36±6 |
| Sweep Into | **74±4** | **74±3** | 67±3 | 63±2 |
| Window Close | **99±0** | 98±1 | 91±2 | 91±3 |

## D. DPPO

Results for DPPO (An et al., 2023), which were omitted due to space constraints and comparatively weak performance, are given below. Meta-World tasks are presented in Table 9 and locomotion tasks in Table 10.

*Table 9.* DPPO task success (%) $\pm$ 2SE.

| Task | 50 | 500 |
|------|-----|-----|
| Button Press | $15 \pm 4$ | $15 \pm 3$ |
| Door Open | $15 \pm 5$ | $15 \pm 6$ |
| Drawer Open | $9 \pm 3$ | $13 \pm 2$ |
| Faucet Close | $22 \pm 5$ | $32 \pm 5$ |
| Lever Pull | $6 \pm 1$ | $4 \pm 1$ |
| Peg Insert Side | $1 \pm 0$ | $1 \pm 0$ |
| Plate Slide | $17 \pm 5$ | $10 \pm 2$ |
| Push Back | $4 \pm 1$ | $3 \pm 1$ |
| Sweep Into | $9 \pm 2$ | $9 \pm 2$ |
| Window Close | $29 \pm 7$ | $35 \pm 3$ |

*Table 10.* DPPO average episode returns $\pm$ 2SE.

| Task | 50 | 500 |
|------|-----|-----|
| HalfCheetah | $-90 \pm 10$ | $-97 \pm 14$ |
| Hopper | $40 \pm 59$ | $74 \pm 31$ |
| Walker2d | $35 \pm 27$ | $32 \pm 13$ |
| Ant | $-554 \pm 28$ | $-424 \pm 22$ |

# E. Preference Dataset Generation

We evaluate on Meta-World environments with the modifications of CPL (Hejna et al., 2024). In contrast to the original Meta-World tasks (Yu et al., 2020), these are modified by randomizing the goal but including the target in the state observation, as well as randomizing the initial robot position and removing the proprioceptive state history from the observation. Additionally, we evaluate our method on four locomotion tasks, namely Ant, HalfCheetah, Hopper, and Walker2d. For all tasks, we train one policy with SAC (Haarnoja et al., 2018) and choose rollouts from four checkpoints throughout training. For the Meta-World tasks, exact step counts, average return, and success rate are given in Table 11. For the locomotion tasks, the relevant values are given in Table 12.

*Table 11.* Dataset quality for the different Meta-World tasks.

| Task | Policy 1 | | | Policy 2 | | | Policy 3 | | | Best Policy | | |
|------|------|--------|---------|------|--------|---------|------|--------|---------|------|--------|---------|
| | Step | Return | Success | Step | Return | Success | Step | Return | Success | Step | Return | Success |
| Button Press | 40k | 509 | 16% | 70k | 1335 | 73% | 150k | 1320 | 74% | 240k | 1709 | 96% |
| Door Open | 30k | 462 | 2% | 50k | 708 | 41% | 70k | 1314 | 83% | 830k | 1994 | 100% |
| Drawer Open | 210k | 1279 | 8% | 270k | 1535 | 61% | 290k | 1640 | 76% | 350k | 1786 | 91% |
| Faucet Close | 30k | 849 | 1% | 60k | 1346 | 36% | 90k | 1451 | 51% | 140k | 2124 | 98% |
| Lever Pull | 30k | 204 | 0% | 190k | 265 | 23% | 300k | 382 | 52% | 640k | 720 | 80% |
| Peg Insert Side | 340k | 735 | 4% | 390k | 1201 | 33% | 410k | 1507 | 86% | 480k | 1760 | 97% |
| Plate Slide | 50k | 487 | 9% | 60k | 448 | 8% | 120k | 1543 | 78% | 250k | 2006 | 99% |
| Push Back | 280k | 47 | 12% | 290k | 102 | 20% | 410k | 573 | 57% | 600k | 1622 | 97% |
| Sweep Into | 50k | 142 | 0% | 150k | 490 | 52% | 300k | 1176 | 81% | 910k | 1958 | 97% |
| Window Close | 30k | 240 | 5% | 70k | 669 | 48% | 80k | 906 | 14% | 120k | 1524 | 99% |

*Table 12.* Dataset quality for the different locomotion tasks.

| Task | Policy 1 | | Policy 2 | | Policy 3 | | Policy 4 | |
| --- | --- | --- | --- | --- | --- | --- | --- | --- |
| | Step | Return | Step | Return | Step | Return | Step | Return |
| Ant | 10k | 245 | 500k | 376 | 1M | 660 | 2M | 935 |
| HalfCheetah | 30k | 76 | 90k | 1071 | 200k | 1583 | 300k | 2499 |
| Hopper | 30k | 339 | 160k | 289 | 260k | 693 | 400k | 788 |
| Walker2d | 60k | 59 | 280k | 260 | 340k | 527 | 500k | 1093 |

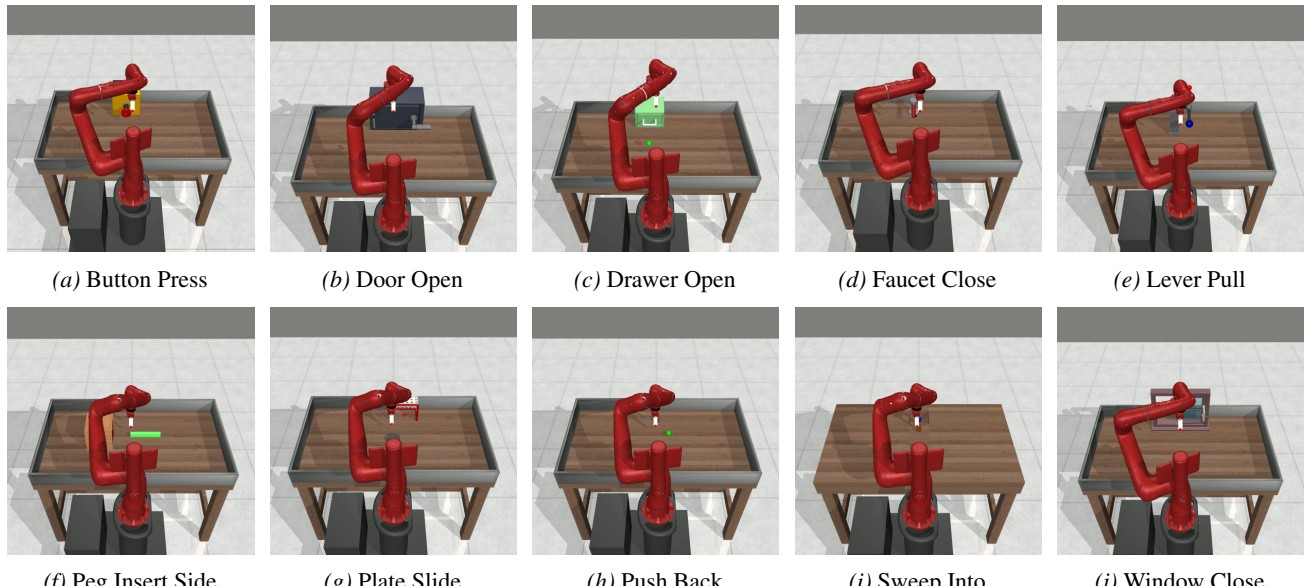

*(a)* Button Press  *(b)* Door Open  *(c)* Drawer Open  *(d)* Faucet Close  *(e)* Lever Pull

*(f)* Peg Insert Side  *(g)* Plate Slide  *(h)* Push Back  *(i)* Sweep Into  *(j)* Window Close

*Figure 4.* Meta-World manipulation tasks used in our experiments. Each panel shows an initial configuration (a–j).

### E.1. Meta-World Task Descriptions

We evaluate on 10 Meta-World manipulation tasks shown in Figure 4: *(a) Button Press* press a small button on the panel until it activates; *(b) Door Open* open a hinged door by pulling the handle to a target angle; *(c) Drawer Open* pull the drawer along its rail until it reaches the goal extension; *(d) Faucet Close* rotate the faucet handle clockwise until it is fully off; *(e) Lever Pull* pull the short lever down through a quarter turn; *(f) Peg Insert Side* insert a cylindrical peg into a horizontal side hole without jamming; *(g) Plate Slide* push the plate across the table into the marked goal region; *(h) Push Back* push the movable puck backward to the target position; *(i) Sweep Into* sweep a small object across the surface into a container opening; *(j) Window Close* push the sliding window pane along its track until fully closed.

### E.2. Locomotion Task Descriptions

We evaluate four different locomotion tasks (Towers et al., 2024), namely Ant, HalfCheetah, Hopper, and Walker2d. All tasks have the same episode length of 250 steps, and episodes terminate only when the final step is reached. There are no early-termination conditions, which are otherwise present in Towers et al. (2024), because this is required for consistent generation of the trajectories used for preferences. The tasks are visualized in Figure 5.

## F. Algorithm Details and Hyperparameters

The method hyperparameters are listed in the following tables: Table 13 for the baselines and Table 14 for our method. Our method is also summarized in Algorithm 1.

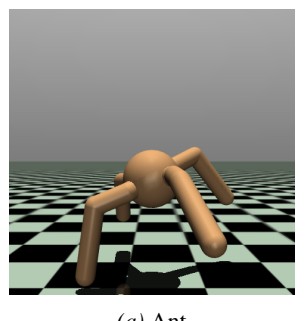 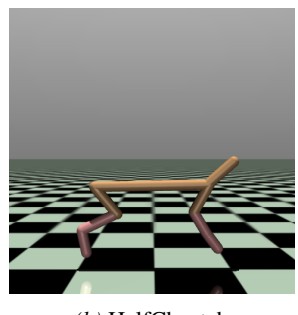 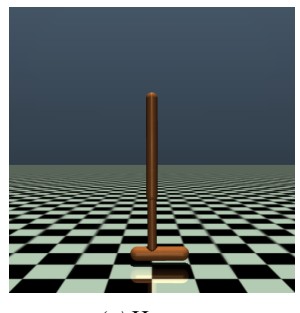 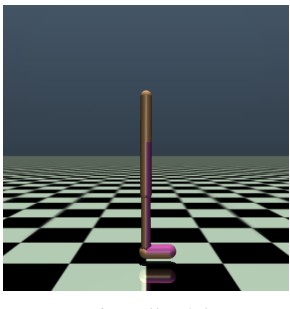

| *(a)* Ant | *(b)* HalfCheetah | *(c)* Hopper | *(d)* Walker2d |

*Figure 5.* Locomotion tasks.

*Table 13.* Hyperparameters for the baselines.

| Hyperparameter | CPL | CPL (KL) | P-IQL | Pref Trans. | IPL |
|---|---|---|---|---|---|
| Learning Rate | 0.0001 | 0.0001 | 0.0003 | 0.0003 | 0.0001 |
| Temp $\alpha$ | 0.1 | 0.2 | - | - | - |
| Bias $\lambda$ | 0.5 | 0.75 | - | - | - |
| $\gamma$ | - | - | 0.99 | 0.99 | 0.99 |
| Expectile $\tau$ | - | - | 0.7 | 0.7 | 0.75 |
| Temperature $\beta$ | - | - | 0.3333 | 3.0 | 0.3333 |
| Hidden Dims | - | - | - | (256, 256) | (512, 512) |
| Soft Target $\tau$ | - | - | - | 0.005 | 0.005 |
| Reward Net Steps | - | - | 50000 | - | - |
| Evaluation Step | 5000 | 5000 | 5000 | 5000 | 5000 |

## G. Ablation: Spearman's Rank Correlation Coefficients

To compute the correlation of the learned model's action likelihoods with the ground-truth advantage, we compute Spearman's correlation coefficient between the log likelihoods of the expert policy of a segment and the log likelihoods coming from our trained policy for the same segment. The Spearman's correlation coefficient of sequences $X_i$ and $Y_i$ first gives each value an integer rank in increasing order and in particular for unique values, it holds that

$$\text{rank}(X) = |\{X_j \leq X\}_j|. \tag{19}$$

Then the sequences $\text{rank}(X_i)$ and $\text{rank}(Y_i)$ are compared using the Pearson correlation coefficient. In contrast to the direct Pearson correlation coefficient, this also measures nonlinear but monotonic correlation between the two sequences. This is relevant for measuring the credit assignment as this removes the impact of the absolute quality of the policy likelihood fit to the expert policy likelihood, that is present in a linear correlation measurement, but measures whether the same state-action pairs of the sequence have high or low likelihood, hence inferred advantage, *within* the segment.

We use the Transformer-based advantage function and evaluate after 10,000 policy update steps. For each of $K = 1{,}000$ segments per task, we determine the rank correlation between the predicted policy likelihoods and the ground-truth expert policy likelihoods, that were used for preference label generation. Practically, we rank the log likelihoods which is equivalent due to the monotonicity of the logarithm. The reported values in Table 15 are averages over the training results of 5 seeds.

*Table 15.* Spearman correlation coefficient between the action likelihoods of the learned policy and the expert policy for each Meta-World task. We compare the **State** versus the **Segment** representation.

| | Button Press | Door Open | Drawer Open | Faucet Close | Lever Pull | Peg Insert Side | Plate Slide | Push Back | Sweep Into | Window Close | Mean |
|---|---|---|---|---|---|---|---|---|---|---|---|
| **State** | 0.06 | 0.11 | -0.34 | 0.35 | **-0.1** | 0.03 | 0.26 | 0.05 | -0.04 | 0.16 | **0.054** |
| **Segment** | **0.36** | **0.21** | **0.03** | **0.57** | -0.26 | **0.24** | **0.27** | **0.20** | **0.00** | **0.55** | **0.217** |

*Table 14.* Hyperparameters for PAWS.

| Hyperparameter | PAWS (MLP) | PAWS (Transformer) |
|---|---|---|
| **Actor networks** | | |
| Learning rate | 0.0003 | 0.0003 |
| Dropout | 0.25 | 0.25 |
| Hidden dimension | 512 | 512 |
| Hidden depth | 2 | 2 |
| Evaluation Step | 250 | 250 |
| Effective sample size $n_{\text{eff}}^*$ (Meta-World / Locomotion) | 10% / 30% | 10% / 30% |
| **Reward networks** | | |
| Learning rate | 0.0003 | 0.0003 |
| Max Len | - | 64 |
| Dropout | - | 0.1 |
| Hidden dimension | 512 | 512 |
| Number of heads | - | 8 |
| Number of layers | 3 | 4 |
| Position encoding | - | learned |
| Min. early stopping value $\alpha$ | 0.995 | 0.995 |

# H. Human Preferences

We collected preferences from 10 non-author human labelers on two Meta-World tasks, Button Press and Door Open. Each labeler provided 50 pairwise comparisons per task from 100 segments, yielding 500 comparisons per task in total. The preference pairs were generated in the same way as in Section E, with the labels provided by humans rather than the oracle. To collect the labels, we used the GUI shown in Figure 6, which presented the two trajectory videos side by side and required the labeler to choose one of them as preferred (no tie option was provided). Each labeler was provided with the task description as defined in Section E.1.

The data collection does not constitute human subjects research in the regulated sense. The participants viewed pairs of robot execution videos and indicated preferences, with no personal data collected, anonymous participation, and no intervention or risk. The object of study is the robot executions; participants served purely as annotators, analogous to crowdsourced labeling. Our institution does not mandate ethics approval for anonymous, minimal-risk annotation of this kind.

To assess statistical significance, we performed pairwise Welch's t-tests comparing each PAWS variant against all baselines. On Door Open, PAWS (MLP) significantly outperforms all baselines, and PAWS (Trans.) significantly outperforms all methods except Preference Transformer. After applying Bonferroni correction ($\alpha = 0.05/24 = 0.0021$, where $24 = 2$ tasks $\times 2$ PAWS variants $\times 6$ baselines), PAWS (MLP) on Door Open remains significant against BC, P-IQL, CPL, CPL+KL, and IPL ($p_{\text{Bonf}} < 0.05$). Under the less conservative Benjamini-Hochberg FDR correction ($q = 0.05$), PAWS (MLP) on Door Open additionally retains significance against Pref Trans. ($p_{\text{BH}} < 0.05$), and both PAWS variants retain significance against BC on Button Press ($p_{\text{BH}} < 0.01$).

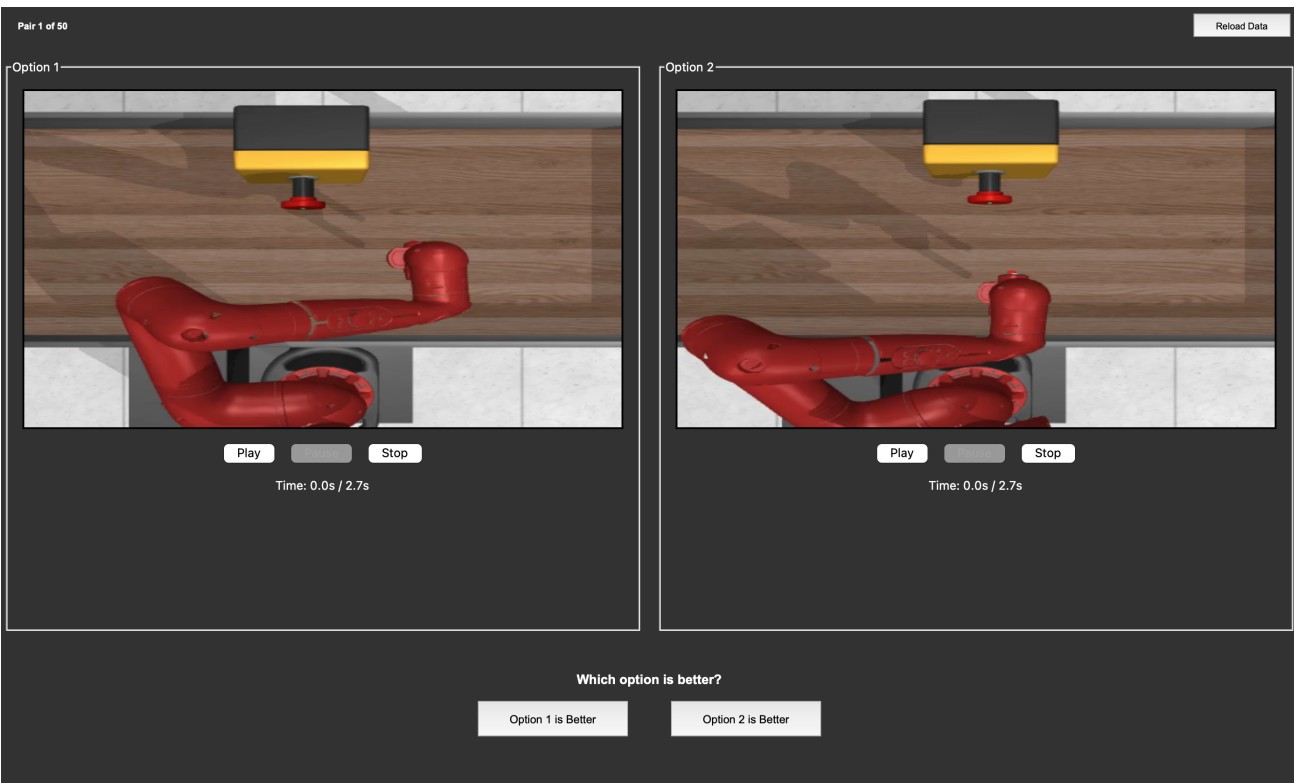

*Figure 6.* GUI for collecting preference labels from humans, shown here for the Button Press task. The labeler sees two robot execution videos side by side, with Play, Pause, and Stop controls for each, and selects which option is better. The header shows progress through the 50 pairs for the current task. The Door Open task uses the same layout with different videos.

---

**Algorithm 1 PAWS: Preference Learning with Segment-Level Advantage Optimization**

---

1: **Input:** Offline segment dataset $\mathcal{D} = \{\tau_i\}_{i=1}^{K}$ sampled from $p_{\mathcal{D}}(\tau)$
2: **Input:** Offline preference dataset $\mathcal{D}_{\text{pref}} = \{(\tau_i^+, \tau_i^-)\}_{i=1}^{n}$, where $\tau_i^+ \succ \tau_i^-$
3: **Input:** Advantage model $A_\phi(s, a)$, policy $\pi_\theta(a \mid s)$
4: Initialize $\phi$ (advantage) and $\theta$ (policy)

   **1. Advantage learning on preference data**
5: **while** advantage parameters $\phi$ are not converged **do**
6:     Minimize preference loss (Equation (2))

$$\mathcal{L}_{\text{pref}}(\phi) = -\frac{1}{n} \sum_{(\tau^+, \tau^-) \in \mathcal{D}_{\text{pref}}} \log \sigma(A_\phi(\tau^+) - A_\phi(\tau^-))$$

7:     Update $\phi \leftarrow \phi - \alpha_\phi \nabla_\phi \mathcal{L}_{\text{pref}}$
8: **end while**

   **2. Lagrange Multiplier Computation**
9: Find $\lambda$ resulting in desired $n_{\text{eff}}^*$ using Equation (10)
10: $\lambda \leftarrow \arg\min_\lambda |n_{\text{eff}}^* - n_{\text{eff}}(\lambda)|$

   **3. Policy update**
11: **while** policy parameters $\theta$ are not converged **do**
12:     **for** each $\tau_i \in \mathcal{D}$ **do**
13:         **Compute importance weights (Equation (7))**
14:         $w_i \leftarrow \exp\left(\frac{1}{\lambda} A_\phi(\tau_i)\right)$
15:         Self-normalize for numerical stability
16:         $w_i \leftarrow \frac{w_i}{\sum_j w_j}$
17:     **end for**
18:     **Policy update using weighted maximum-likelihood (Equation (8))**
19:     Compute gradient:

$$\nabla_\theta \mathcal{L} = \frac{1}{K} \sum_i w_i \sum_t \nabla_\theta \log \pi_\theta(a_t^i \mid s_t^i)$$

20:     Update $\theta \leftarrow \theta + \alpha_\theta \nabla_\theta \mathcal{L}$
21: **end while**
22: **return** $\pi_\theta$

---

