# OpenReview forum: "PAWS: Preference Learning with Advantage-Weighted Segments"
_ICML.cc/2026/Conference — ICML 2026 regular_

### Official Review · Reviewer_YiX1 · 2026-03-04

**Soundness:** 3
**Presentation:** 2
**Significance:** 2
**Originality:** 3
**Overall Recommendation:** 4
**Confidence:** 3

**Summary:**

This paper presents PAWS, a method for preference-based reinforcement learning (PbRL) that aims to address the temporal credit assignment problem. The approach first learns an advantage function defined over trajectory segments and then improves the policy using advantage-weighted optimization over these segments. The authors evaluate the method on a range of simulated tasks and report performance gains over several existing baselines, under both simulated and real human feedback.

**Compliance With Llm Reviewing Policy:**

Affirmed.

**Final Justification:**

This paper presents PAWS, a method for preference-based reinforcement learning (PbRL) that aims to address the temporal credit assignment problem. Although the initial version has some presentation issues, the authors are able to address them in the rebuttal. Thus I raise the score from 3 to 4 (weak accept).

**Key Questions For Authors:**

1. Why using a per-step advantage estimator can address the problem of credit assignment?
2. How can we reach eq. 10? And why can we control Lagrangian multiplier by tuning $n_{eff}$? And actually how can $n_{eff}$ be implemented?
3. Why using expert policy to generate preference labels given the fact that reward functions are available? Will it make any difference if ground truth reward is used for preference labels?
4. (Not necessary) Is the learned advantage compatible with policy gradient methods like PPO?

**Limitations:**

The limitation of lack of theoretical explanation link between advantage learning and credit assignment should be listed, or addressed in paper.

**Strengths And Weaknesses:**

## Strengths
1. The paper is well structured.
2. The paper proposes theoretical justification to support the proposed advantage learning algorithm.
3. The experiments are comprehensive, including both simulated and real-human feedback.

## Weaknesses
1. It is not fully convincing that the method is conceptually distinct from standard reward learning. In the current formulation, the advantage function appears to be learned in essentially the same way as a reward model and is later used in a role very similar to a reward function for policy optimization. The paper does not clearly articulate what fundamentally distinguishes “advantage learning” from simply learning a reward.
2. The claimed link between advantage learning and credit assignment is not clearly justified. Although the paper suggests that learning an advantage function can improve temporal credit assignment, it does not provide a clear theoretical explanation or formal analysis to support this claim. In particular, the authors refer to “segment-wise advantage estimation,” but in Section III the advantage function is parameterized and learned in a step-wise manner. This mismatch between the conceptual description and the actual implementation makes the argument less convincing and raises questions about how the method truly addresses credit assignment.
3. Section 3.3 is particularly difficult to understand. It is very hard to follow the notations. For example, there is no explanation why $n_{eff}$ is the desired number of effective samples, and how can we calculate $\epsilon$ from $n_{eff}$.
4. In experiments, the reason of using expert policy to generate preference labels is unclear.

---

> ### Author Rebuttal · Authors · 2026-03-31
>
> We thank the reviewer for the constructive review and for acknowledging the theoretical justification and experimental evaluation. Below we provide answers to the Weaknesses (W) and Questions (Q).
>
> **W1:** Recent work by [1] shows that human preferences are better modeled using the advantage function, not the reward function. The standard reward learning approach based on partial returns is not a good model of human preferences, and what is actually being learned corresponds to the advantage function. The advantage function captures relative action quality at a given state, whereas a reward function assigns absolute credit to transitions. Based on this insight, we developed our method.
>
> **W2:** In our method, updates on segments alleviate the temporal credit assignment problem. The advantage function is defined on individual state-action pairs but learned using segment-level preferences. We aggregate advantage values over whole segments and perform policy updates at the segment level, for which we provide theoretical justification. We will update Section 3 to clarify this.
>
> **W3:** The effective sample size (ESS) [2] (Eq. 10) is a common importance sampling metric [3]. It measures how many samples meaningfully contribute to a weighted estimate. In PAWS, we define a desired $n^{*}_{eff}$ to achieve.
>
> How $\\epsilon$ is obtained from $n^{*}_{eff}$: all quantities are coupled through $\\lambda$:
> 1. **$\\epsilon$ determines $\\lambda$** via dual function minimization (Proposition 3.2, Eq. 9).
> 2. **$\\lambda$ determines the weights** $w_{i} = \\exp(A_{\\phi}(\\tau_{i})/\\lambda)$ (Eq. 7).
> 3. **The weights determine $n_{eff}$** via Eq. 10.
>
> Rather than tuning $\\epsilon$ (which depends on action dimensionality), we specify  $n_{eff}$ and search for $\\lambda$ that yields this target. In practice, this is a one-dimensional search: $\\lambda \\leftarrow \\arg\\min\{\\lambda} |n^*_{\\text{eff}} - n_{\\text{eff}}(\\lambda)|$ (Algorithm 1, line 10, Appendix F).
>
> **W4:** On page 6 (line 297, left column) we provide justification, but agree it may be insufficient. Below we elaborate, and will update the final version.
>
> We used log-likelihoods of a SAC-trained policy [4] for labeling. In maximum entropy RL, the optimal policy is [5]:
>
> $$\\pi^{*}(a_{t}|s_{t}) = \\exp\\left(\\frac{1}{\\alpha}(Q(s_{t}, a_{t}) - V(s_{t}))\\right)$$
>
> where $Q(s_{t}, a_{t})$ is the soft Q-function, $V(s_{t})$ the soft value function, and $\\alpha$ a scaling parameter. Since $A(s_{t}, a_{t}) = Q(s_{t}, a_{t}) - V(s_{t})$, the log of the optimal policy yields the scaled advantage. This motivates labeling preferences using SAC log-likelihoods.
>
> **Q1:** We do not use a per-step advantage estimator. PAWS uses segment-based updates. See W2 for details.
>
> **Q2:** From Eq. 7: $w_{i} = \\exp\\left(\\frac{1}{\\lambda} \\sum_{t} A(s_{t}^{i}, a_{t}^{i})\\right)$. The ESS [2] is $n_{eff} = (\\sum_{i} w_{i})^{2} / (\\sum_{i} w_{i}^{2})$, computed under a given $\\lambda$. Note $\\lambda$ depends on $\\epsilon$ via Eq. 9.
>
> To achieve a desired $n_{eff}$, we perform a line search over $\\lambda$ (Algorithm 1, lines 9-10, page 20). We will reference the algorithm in the main text to improve readability.
>
> **Q3:** Environment rewards do not represent real human preferences well [1], so using them would not faithfully represent human labelers. See W4 for our labeling justification. All baselines use the same labels, ensuring a fair comparison.
>
> **Q4:** The advantage function is in principle compatible with PPO. However, in the offline setting, sampling out-of-distribution actions risks unreliable estimates. Extending PAWS to online settings is an interesting future direction.
>
> **References:**
>
> [1] Knox et al. (2024). Models of human preference for learning reward functions. TMLR.
>
> [2] Martino, Elvira, & Louzada (2017). Effective sample size for importance sampling. Signal Processing.
>
> [3] Owen (2013). Monte Carlo theory, methods and examples.
>
> [4] Haarnoja et al. (2018). Soft actor-critic. ICML.
>
> [5] Haarnoja et al. (2017). RL with deep energy-based policies. ICML.

---

> > ### Author Rebuttal · Reviewer_YiX1 · 2026-04-01
> >
> > Thanks the authors for their effort in rebuttal. The authors addressed all of my proposed weakness and problems. I will raise the score from 3 to 4.

---

> > > ### Author Response · Authors · 2026-04-08
> > >
> > > We sincerely thank the reviewer for the thoughtful engagement with our rebuttal and for raising the score. We are glad that our responses fully resolved the concerns, and we will incorporate all clarifications discussed during the rebuttal into the final version of the paper. We appreciate the reviewer's constructive feedback, which has helped us improve our work.

---

### Official Review · Reviewer_AzhB · 2026-03-08

**Soundness:** 4
**Presentation:** 4
**Significance:** 2
**Originality:** 2
**Overall Recommendation:** 4
**Confidence:** 4

**Summary:**

This paper identifies a problem relating to the mismatch in training advantage functions on trajectory-segment-level preferences, but then querying the learned advantage functions on single transitions during inference time. The authors instead propose optimizing the policy by performing updates directly on trajectory segments; the proposed method is novel in how the policy is derived from the advantage function, where trajectory segments are weighted by their advantage during policy learning. The authors show their proposed method, PAWS, outperforms a number of other PbRL methods across a suite of robotics environments. Their method also exhibits robustness to increasing the trajectory-segment length that preferences are elicited over, and they empirically show that training a policy with the learned advantage function and trajectory-segment level updates (i.e., their proposed method) outperforms using single-step updates from the same advantage function.

**Compliance With Llm Reviewing Policy:**

Affirmed.

**Final Justification:**

I maintain my positive score; it is not higher because my main concerns around novelty still apply.

**Key Questions For Authors:**

How sensitive is PAWS to errors in the learned advantage estimates, particularly when preferences are noisy or inconsistent across trajectory segments?

**Limitations:**

yes

**Strengths And Weaknesses:**

The proposed problem and solution are elegant and sensible. I view the main weakness of this work as the significance of its contribution; the policy optimization method the authors propose seems like a small change from prior work. I do appreciate, however, that this subtle change addresses an important problem and empirically induces substantially better performance than a suite of PbRL baselines.

Soundness
The paper’s methodology is sound, building on established prior work with strong theoretical (and intuitive) backing.

Presentation
The paper’s presentation is clear.

Significance/Originality
My main concern is regarding the papers significance and originality. The authors identify a novel problem—the mismatch in training advantage functions on trajectory-segment-level preferences but sampling the learned advantage function for individual transitions. But the authors’ proposed solution to this problem feels incremental given prior work in policy optimization.

---

> ### Author Rebuttal · Authors · 2026-03-31
>
> We sincerely thank the reviewer for the positive and encouraging assessment of our work, and for recognizing the soundness of our methodology, the clarity of presentation, and the substantial empirical improvements.
>
> The reviewer raises as the main concern that the work's significance and originality are limited. However, as the reviewer already pointed out, we identified a novel problem and proposed an elegant and sensible method to address it. We agree that the method builds on existing optimization techniques, but we have adapted them to the preference learning setting in a non-trivial way. In particular, we derive the segment-based policy updates and introduce the effective sample size mechanism for more intuitive hyperparameter setting, which is novel in this context. Furthermore, we believe that clearly diagnosing and analyzing this problem is itself a valuable contribution that can inspire future research to develop alternative and potentially more sophisticated solutions. Our work provides both a concrete baseline and diagnostic tools (e.g., the Spearman rank correlation analysis and segment length ablations) that future methods can build upon. Additionally, we see the methodological simplicity of PAWS as a practical strength: it delivers consistent improvements across 14 tasks with minimal added complexity, making it straightforward to adopt.
>
> > How sensitive is PAWS to errors in the learned advantage estimates, particularly when preferences are noisy or inconsistent across trajectory segments?
> >
>
> To address this question, we performed an additional evaluation with preferences collected from human participants. Humans are inherently noisy and inconsistent, making this a natural test of robustness to imperfect preferences.
>
> ### Evaluation with Extended Human-collected Preferences
> In the paper, we originally reported results based on preferences from a single participant. To provide a more representative evaluation, we subsequently gathered preferences from five additional participants. The mean success rates (%) are presented below, and our method continues to outperform all baseline approaches.
>
> | Task | BC | P-IQL | CPL | CPL+KL | Pref Trans. | IPL | PAWS (Trans.) | PAWS (MLP) |
> | --- | --- | --- | --- | --- | --- | --- | --- | --- |
> | Button Press | 44 | 52 | 50 | 57 | 61 | 50 | **67** | 66 |
> | Door Open | 71 | 66 | 58 | 66 | 77 | 76 | **90** | 77 |
>
> **References:**
>
> [1] Hejna et al. (2024). Contrastive Preference Learning. ICLR.
>
> [2] Knox et al. (2024). Models of human preference for learning reward functions. TMLR.

---

> > ### Author Rebuttal · Reviewer_AzhB · 2026-04-01
> >
> > I appreciate your additional human studies, but my main concerns around novelty still apply. I will keep my score as is.

---

> > > ### Author Response · Authors · 2026-04-07
> > >
> > > We thank the reviewer for engaging with our response and for maintaining their positive recommendation. We would like to briefly clarify why we believe the novelty of PAWS may be undervalued.
> > >
> > > The contribution of PAWS lies in the combination of three points that together, in our view, go beyond an incremental change:
> > >
> > > 1. **Problem identification**. We identify and analyze the segment-to-step mismatch in PbRL as a training and inference distribution shift, a perspective that motivates concrete diagnostics and predictions, which we validate empirically. The reviewer themselves kindly characterized this as a novel problem.
> > > 2. **A preference-specific adaptation of advantage-weighted policy updates**. Advantage-weighted policy updates are not new, but to our knowledge, no prior work learns the advantage function directly from segment-level preferences without a separate reward model and applies the weighted update at the segment level so that training and inference distributions remain aligned.
> > > 3. **Effective-sample-size-based trust region control for PbRL**. The $n_{eff}$ mechanism replaces the opaque KL bound $\epsilon$ with a quantity practitioners can reason about directly, and is, to our knowledge, new in the PbRL context.
> > >
> > > We would also like to highlight that our empirical results are consistent across 14 tasks in both low- and high-data regimes, and are supported by extensive ablations (on segment length, effective sample size, advantage architecture, and segment- vs. step-level updates) that directly substantiate the claims underlying our method. During the discussion period, we further expanded our human-preference evaluation with an even larger user study, and PAWS continues to outperform all baselines under this realistic, noisy supervision. Beyond the method itself, we hope that the diagnosis and analysis we provide, contribute to the field by drawing attention to this research direction and enabling future work to build on them.
> > >
> > > We thank the reviewer once more for their careful reading and constructive engagement.

---

### Official Review · Reviewer_7zrZ · 2026-03-10

**Soundness:** 2
**Presentation:** 2
**Significance:** 3
**Originality:** 3
**Overall Recommendation:** 4
**Confidence:** 3

**Summary:**

This paper focuses on the credit assignment problem in preference based RL (PbRL). In most PbRL works, it is common to gather preferences over trajectories and use these to learn a reward function. However, there could be several predicted returns that preserve a preference; there could be issues when using the reward function at the state, action level. In particular, the authors describe a mismatch betwen segment level training and step level inference. To address this mismatch, PAWS is introduced. This method updates the policy at the segment level by using an advantage function over segments.

**Compliance With Llm Reviewing Policy:**

Affirmed.

**Final Justification:**

The authors have addressed my main concerns in the rebuttal. They have included a rigorous evaluation of their method.

**Key Questions For Authors:**

1. In the results section, under Meta-World tasks, it says, "Specifically, we execute 25 rollouts per evaluation point, smooth the results by averaging over eight adjacent points, and report the peak performance". What does this mean? What are the adjacent points? It's not clear what is being reported. Also, is this a normal way to aggregate results? If not, why did the authors choose this method?
2. For the human experiments, were these preferences from a single human or multiple? Was this person not part of the author list? Relatedly, why did the authors not use the ground truth return to label the trajectories? This is commonly done in most PbRL evaluations.
3. Can the authors explain the connection between the use of the advantage function and alleviating the credit assignment problem?

**Limitations:**

The authors did include limitations of their work; however, one limitation that is missing is the lack of more thorough human subject studies. From how it is written in the paper, it seems only 1 human provided preferences. This is not enough to make any claims.

**Strengths And Weaknesses:**

Soundness:
- The paper does extensive experiments in Meta-World and Mujoco environments, showing mean + 2*standard error. It does include some human-labeled experiments, but the lack of details on the human study is concerning.
- This paper focuses on temporal credit assignment. However, it is not clear how the method specifically addresses this.
- In addition, the authors state, "The mismatch between segment-level training and step-level inference induces a distribution shift that is largely overlooked in existing PbRL approaches." Aside from the toy example in Figure 2, is there a way of verifying that this issue is arising in the experiments?
- The authors also note that other baselines that do not learn reward models "struggle with tasks that require complex reasoning (Ivison et al., 2024; Xu et al., 2024) and in settings with limited preference data." Since this proposed method is also not learning a reward model, from my understanding, I'm unsure why these concerns are not valid for PAWS.
- The authors state, "As can be seen, state-action-based updates suffer a noticeable performance degradation as segment length increases, reflecting the ever-larger severity of the training and inference distribution shift", regarding Figure 3b. But it seems like segment sizes 16, 32, and 64 all have about the same success rate, so I do not think this claim is true based on this result.



Presentation
- The authors repeated the paragraph "Segments and State-action based updates" twice on page 7.
- The figures are very visually appealing. I also like the idea behind the coloring of the tables, but it would be nice if the authors described in the text how many times their proposed method was significantly different from the other baselines (using statistical tests). This would make it easier to understand, as the Tables (particularly Table 1) have a lot of numbers to parse.

Significance:
- Credit assignment is a long-standing problem in RL. I think many people in the community would value this type of work.

Originality:
- This is the first time I have seen PbRL focusing on the trajectory level (e.g., not just learning a reward function from trajectories but then using it at the s,a level).
- There are a few propositions included in the text, but I am unsure if these are new or were from Peters et al. (2010).


References:

Peters, J., Mulling, K., and Altun, Y. Relative entropy
policy search. In Proceedings of the AAAI Conference on
Artificial Intelligence, volume 24, pp. 1607–1612, 2010.

---

> ### Author Rebuttal · Authors · 2026-03-31
>
> We thank the reviewer for their detailed feedback and for engaging with our work. Below we address the points raised.
>
> > The lack of details on the human study is concerning.
>
> We will update the manuscript with details about the human study. In the paper, we collected preferences from one participant. We have now collected preferences from 5 additional humans. Results are in our response to Reviewer AzhB and indicate that PAWS remains superior to the baselines. We are happy to provide any additional details the reviewer may find helpful.
>
> > It is not clear how the method specifically addresses [temporal credit assignment]. Aside from the toy example in Figure 2, is there a way of verifying that this issue is arising in the experiments?
>
> We provide experimental evidence beyond Figure 2. The Spearman's rank correlation analysis (Page 8) compares segment-based and state-based updates: with step-based updates, the mean correlation is only 0.054, indicating policy likelihoods are essentially uncorrelated with expert likelihoods. Segment-based updates achieve 0.217. Table 4 further shows segment-based updates consistently outperform state-based updates using the same advantage function. We elaborate in Q3.
>
> > Since this proposed method is also not learning a reward model, I'm unsure why these concerns are not valid for PAWS.
>
> While PAWS does not learn a reward model, it learns an advantage function, a utility model that captures dependencies across state-action pairs within segments. This distinguishes it from direct preference methods like DPO or CPL [1], which optimize policy likelihoods without any intermediate utility model.
>
> > It seems like segment sizes 16, 32, and 64 all have about the same success rate, so I do not think this claim is true.
>
> The reviewer references Figure 3b, but the quoted sentence refers to Figure 3c. We agree the drop is most pronounced from segment length 1 to 16 and does not increase substantially beyond that. We will clarify this to avoid overstatement.
>
> > It would be nice if the authors described how many times their proposed method was significantly different from baselines (using statistical tests).
>
> With 500 preferences, PAWS achieves significant improvements (p<0.05, z-test) over BC on all 10 tasks, over IPL on 9/10, over CPL, CPL+KL, and P-IQL on 8/10, and Pref Trans. on 7/10 tasks. In non-significant comparisons, PAWS was on par, and no baseline is significantly better. With 50 preferences, results are task-dependent with no clear winners. We will include this in the revision.
>
> > I am unsure if [the propositions] are new or were from Peters et al. (2010).
>
> The propositions are derived from Peters et al. (2010).
>
> Below are the answers to the questions.
>
> **Q1:** We follow the evaluation scheme of CPL [1]. The policy is evaluated at regular intervals. A running average over eight consecutive checkpoints is computed, and the peak of this smoothed curve is reported.
>
> **Q2:** In the original submission, preferences were collected by one author. We have since collected from 5 additional people, 2 of whom are authors. We will discuss limitations in the revision. Results are in our response to Reviewer AzhB. Regarding ground-truth rewards: recent work [2] shows they do not model real human preferences well, and newer methods [1] have moved away from them.
>
> **Q3:** We use the advantage function because it better represents human preferences, as shown in [2]. PAWS updates the policy on segments, which alleviates the credit assignment problem as described in Section 3, supported by Table 4, Figure 3c, and the Spearman's rank correlation analysis. The advantage function alone does not resolve credit assignment; the segment-based updates do. We additionally evaluated PAWS with shorter update segments (32, 16, 8) than the training length of 64, and those results further support our claim. Details are in our response to Q1 of Reviewer UPXy.
>
> **References:**
>
> [1] Hejna et al. (2024). Contrastive Preference Learning. ICLR.
>
> [2] Knox et al. (2024). Models of human preference for learning reward functions. TMLR.

---

> > ### Author Rebuttal · Reviewer_7zrZ · 2026-03-31
> >
> > I thank the authors for their effort in this rebuttal!
> >
> > **Human feedback experiments:**
> >
> > Unfortunately, my concerns with the human studies are not resolved. 2 out of 5 of the participants in the new human data are the authors. I strongly advise against authors participating in their own human study, as this could potentially introduce bias. I am also wondering whether the collection of human data was approved by an external ethics committee. Also, in the actual results from this (under rebuttal for Reviewer AzhB), the authors did not include any sort of variance measure. Are these differences statistically significant?
> >
> > **Statistical testing follow-up questions:**
> >
> > 1. Why did the authors use a z-test? as compared to a t-test (which is commonly used for small sample sizes and when the population variance is unknown).
> > 2. Did the authors check whether the assumptions were met for the z-test (e.g., normality?)
> > 3. Did the authors do a z-test between PAWS and each baseline? If so, the authors should use some sort of family wise error rate correction  (e.g., Bonferroni). Because the more statistical tests performed, the more likely it is that you make a Type 1 error (false positive).
> >
> > **Propositions**:
> >
> > I think the authors should make it clearer that the theory is largely extended from Peters et al. (2010).
> >
> > **Does the method specifically address temporal credit assignment?**
> >
> > The spearman rho correlation for segment-based updates is only 0.217. This is still considered low. So it is still unclear to me how the current results demonstrate that PAWS addresses credit assignment.

---

> > > ### Author Response · Authors · 2026-04-07
> > >
> > > We thank the reviewer for the valuable feedback. Below we address the follow-up questions.
> > >
> > > **Human feedback experiments**
> > >
> > > Due to rebuttal time constraints, we initially collected preferences from authors. We would like to note that author-generated data have been used exclusively in [1,2] and for some tasks in [3]. Regarding ethics approval: the data collection does not constitute human subjects research in the regulated sense. Participants viewed pairs of robot execution videos and indicated preferences. No personal data was collected, participation was anonymous, and there was no intervention or risk. The object of study is the robot executions; humans served purely as annotators, analogous to crowdsourced labeling. None of the related works [1,2,3] that collected human preferences have reported an ethical approval in their papers. Moreover, our institution does not mandate ethical approval for such anonymous, minimal-risk annotation.
> > >
> > > We have collected an entirely new preference dataset from 10 non-author participants. Success rates [%] ± 2SE:
> > >
> > > |  | BC | P-IQL | CPL | CPL+KL | Pref Trans. | IPL | PAWS (Trans.) | PAWS (MLP) |
> > > | --- | --- | --- | --- | --- | --- | --- | --- | --- |
> > > | button press | 44.7±2.4 | 53.2±3.7 | 47.5±7.4 | 49.0±10.1 | 55.2±5.6 | 47.8±4.2 | **57.2±5.6** | 56.2±5.6 |
> > > | door open | 73.6±2.4 | 64.0±3.7 | 55.6±9.2 | 54.8±8.0 | 76.9±4.4 | 76.2±3.3 | 82.2±3.8 | **86.0±3.0** |
> > >
> > > We performed pairwise Welch's t-tests comparing each PAWS variant against all baselines. Uncorrected, on button press both PAWS variants significantly outperform BC and IPL, but do not show significant differences against others. On door open, PAWS (Trans.) significantly outperforms all methods except Pref Trans., and PAWS (MLP) achieves the best performance and significantly outperforms all baselines.
> > >
> > > After applying Bonferroni correction (α=0.05/24=0.0021, where 24 = 2 tasks × 2 PAWS variants × 6 baselines), PAWS (MLP) on door open remains significant against BC, P-IQL, CPL, CPL+KL, and IPL (all p_Bonf<0.05). Under the less conservative Benjamini-Hochberg FDR correction (q=0.05), PAWS (MLP) additionally retains significance against Pref Trans. (p_BH<0.05), and all button press comparisons against BC remain significant for both PAWS variants (p_BH<0.01).
> > >
> > > **Statistical testing**
> > >
> > > We note that no baseline from our paper applies formal significance tests or multiple-comparison corrections in their evaluations. We have nonetheless revised our analysis:
> > >
> > > 1. **Z-test vs. t-test.** We replaced all z-tests with Welch's t-tests; qualitative conclusions are unchanged.
> > > 2. **Normality.** Besides Welch's t-test, which is robust to moderate departures from normality even at small sample sizes (K=10), we additionally verified all conclusions under the non-parametric Wilcoxon rank-sum test; qualitative results are unchanged.
> > > 3. **Multiple comparisons.** With 6 baselines × 10 tasks = 60 tests, we apply Benjamini-Hochberg (FDR q=0.05): 48/60 comparisons remain significant in PAWS's favor, and no baseline is significantly better than PAWS on any task. We chose BH because it controls the false discovery rate, which is appropriate for reporting across multiple tasks where we expect multiple true effects, rather than the more conservative family-wise error rate controlled by Bonferroni; even under Bonferroni, 37/60 remain significant and aggregate conclusions are unaffected.
> > >
> > > **Propositions**
> > >
> > > We will make this clearer. The citation currently appears in the proposition title, but we will make it more prominent in the main text. Our derivation extends Peters et al. (2010) by adapting it to segment-wise updates in an offline setting.
> > >
> > > **Does the method specifically address temporal credit assignment?**
> > >
> > > To clarify, we claim that PAWS alleviates rather than solves temporal credit assignment. In preference learning, the problem cannot be fully solved, as preference data only provides segment-level labels. While the Spearman correlation of 0.217 is modest in absolute terms, it is a significant improvement over the step-based baseline (0.054), demonstrating that segment-based updates meaningfully improve credit assignment despite the inherent difficulty. Beyond the Spearman correlation, our claim is supported by Figure 3c, Table 4, and the additional results provided to Reviewer UPXy (Q1).
> > >
> > > [1] Kim et al. (2023). Preference Transformer: Modeling Human Preferences using Transformers for Reinforcement Learning. ICLR.
> > >
> > > [2] Lee et al. (2021). PEBBLE: Feedback-Efficient Interactive Reinforcement Learning via Relabeling Experience and Unsupervised Pre-training. ICML.
> > >
> > > [3] Christiano et al. (2017). Deep Reinforcement Learning from Human Preferences. NeurIPS.

---

### Official Review · Reviewer_UPXy · 2026-03-12

**Soundness:** 3
**Presentation:** 3
**Significance:** 2
**Originality:** 2
**Overall Recommendation:** 4
**Confidence:** 4

**Summary:**

The distribution shift between utility function and policy optimization is the focus of this paper. Additionally, it offers a novel approach that uses the learned reward to query this function step-by-step in order to obtain the learning signal. Additionally, it assesses the method on meta-world manipulation tasks and d4rl locomation tasks.

**Compliance With Llm Reviewing Policy:**

Affirmed.

**Final Justification:**

The rebuttal addressed two of my initial concerns. The additional human-label experiments from five participants (W2) provide evidence that PAWS handles noisy preferences, and the clarification on oracle labeling (Q2) confirms that all baselines use the same labels, making the comparison controlled. On Q3, the use of a single n_eff=10% across all 10 tasks and both preference budgets in Table 1, without per-task tuning, reduces the concern about hyperparameter sensitivity.

On Q1, partial disagreement remains. The authors' position is that performance degradation with shorter segments supports their core thesis rather than indicating a limitation. This is consistent with the paper's framing, and the comparison between segment-level and state-action-level updates in Table 4 provides empirical support. That said, the method assigns identical importance weights to all state-action pairs within a segment, which relies on the assumption that action quality is temporally correlated. The paper would benefit from a more explicit discussion of when this assumption may not hold and what the consequences would be.

Overall, the paper identifies a real and previously underexplored issue (training-inference distribution shift in PbRL) and proposes a coherent solution through segment-level policy updates. The optimization framework draws on established techniques (Peters et al., 2010), which limits the theoretical novelty, but the adaptation to the preference learning setting is well-executed and supported by experiments across 14 tasks, two data regimes, and multiple ablations. Taking the rebuttal into account, I raise my score from 3 to 4.

**Key Questions For Authors:**

Q1: Every state-action pair in a segment has the same importance weight $\exp(A_\phi(\tau)/\lambda)$ in the segment-level policy update (eq 8). This means that bad actions are upweighted alongside good ones in lengthy segments that contain both high-quality and low-quality actions. Have you assessed PAWS's sensitivity to variations in within-segment quality?

Q2: The log-probabilities of the optimal SAC policy are used by the oracle labeling scheme to determine preferences. This indicates that the preference labels are entirely consistent with PAWS's advantage-based Bradley-Terry model. Does this alignment contribute to the strong performance?

Q3: The relative proportion of the effective sample size, n_eff, is fixed. Figure 3's ablation implies that the ideal n_eff is dependent on the quantity of data available, with larger datasets favoring smaller n_eff. Have you given this a try? The practical benefit of n_eff over direct epsilon-tuning is reduced if the method is sensitive to this decision.

**Limitations:**

yes

**Strengths And Weaknesses:**

Strength: PAWS is an easy-to-understand method. This paper adapts the well-established segment-level trust-region optimization to the offline preference setting.

Weakness: The theoretical contribution is restricted to rephrasing established findings in a novel setting. Standard results from earlier papers are presented in propositions 3.1 and 3.2. The majority of trials use oracle labels, while some use human labels.It's unclear if noise labels make the distribution shift issue worse or better.

---

> ### Author Rebuttal · Authors · 2026-03-31
>
> We thank the reviewer for their insightful comments and deeply appreciate the time and effort dedicated to evaluating our work. Below, we address the mentioned weaknesses and then answer the questions.
>
> > The theoretical contribution is restricted to rephrasing established findings in a novel setting. Standard results from earlier papers are presented in propositions 3.1 and 3.2.
> >
>
> As also noted by reviewer 7zrZ, our approach addresses a significant problem in a novel way by combining and adapting existing methods to the preference learning setting with segment-based updates. The contribution is methodological rather than purely theoretical.
>
> > The majority of trials use oracle labels, while some use human labels. It's unclear if noise labels make the distribution shift issue worse or better.
> >
>
> In the paper, we presented results with preferences collected by one human (Table 3). During the rebuttal period, we collected preferences from 5 additional participants, as humans inherently provide noisy preferences. Due to space constraints, these results are presented in our response to reviewer AzhB. The results show that our method handles noisy data well and remains superior to the baselines.
>
> **Q1:** In our scenario, we assume that action quality is correlated over time, which is reflected by our choice of having different "experts" with varying quality levels for data collection. We believe this is a more realistic assumption than datasets that add Gaussian noise on individual steps to account for varying data quality [1]. Uncorrelated noise would yield highly variable within-segment quality, where single-action importance weights would likely perform better. Yet, as our experiments illustrate, segment-level weights clearly outperform single-step weights in the multi-expert scenario. Additionally, we trained our transformer-based advantage function using 500 preferences with segment length 64, then updated the policy with shorter segments (32, 16, 8) over 5 seeds across all 10 tasks. Results show a performance drop with decreasing segment length on average:
>
> | Success Rate [%] | 64 (Table 1) | 32 | 16 | 8 |
> | --- | --- | --- | --- | --- |
> | Avg. 10 Tasks | 78.2 | 75.8 | 69.9 | 64.3 |
>
> **Q2:** We use the same oracle labels for all baselines, ensuring a controlled comparison. The motivation for using log-probabilities is that they better model human preferences, as shown by [2]. This work demonstrates that reward-based preference modeling is a poor fit for human judgments and has been replaced by advantage-based formulations in recent works such as [1]. Our additional experiments with human-generated labels (see W2 above) further support this claim. We also provide additional supporting arguments in our response W3 to reviewer YiX1.
>
> **Q3:** The value of $n_{\text{eff}}$ depends on the quality and quantity of data. In Figure 3, we test different $n_{\text{eff}}$ values, including 5%, which is lower than our main evaluation in Table 1. Figure 3a evaluates with 500 preferences and 3b with 50, covering different data regimes. The results indicate that with less available data, a higher relative proportion is needed, because overly small $n_{\text{eff}}$ concentrates updates on too few data points, harming generalization. Crucially, $n_{\text{eff}}$ remains more interpretable than $\epsilon$: it directly corresponds to the number of samples effectively contributing to the policy update, which can be estimated based on dataset characteristics. In contrast, $\epsilon$ defines an abstract bound on the KL divergence between distributions, whose appropriate value depends on parameters like action dimensionality and is harder to set in practice.
>
> **References:**
>
> [1] Hejna et al. (2024). Contrastive Preference Learning. ICLR.
>
> [2] Knox et al. (2024). Models of human preference for learning reward functions. TMLR.

---

> > ### Author Rebuttal · Reviewer_UPXy · 2026-04-02
> >
> > Thank you for the response and the additional experiments. The new human data (W2) and explanations (Q2) resolved my concerns about the evaluation setup.
> >
> > However, my core technical concerns remain:
> > - On Q1: Your new results show that performance drops significantly when segments are shorter. This actually confirms my worry: the method cannot evaluate individual actions well and just relies on long segments to smooth out errors.
> > - On Q3: You admitted that $n_{eff}$ is sensitive to data size. Even if it is easy to understand, it still requires careful tuning in practice.

---

> > > ### Author Response · Authors · 2026-04-07
> > >
> > > We thank the reviewer for the follow-up and are glad that concerns W2 and Q2 have been resolved. Regarding the remaining points:
> > >
> > > Q1: We would like to clarify one of our main contributions, as we believe it reframes this concern. The performance drop with shorter segments is precisely the point of our work, not a weakness. A key insight of our paper is that state utility functions learned from preferences cannot reliably assign credit to individual actions, which is why segment-level updates are needed. The shorter-segment results therefore support our contribution rather than undermine it: they provide additional empirical evidence that policy updates should be performed over whole segments.
> > >
> > > The reviewer frames segment-level updates as a limitation ("the method cannot evaluate individual actions well"), but our paper explicitly argues that evaluating individual actions from preference data is the wrong objective in the first place, because preferences are always provided over segments. Segment-level updates are the appropriate response to this difficulty, not a workaround for a flaw in PAWS. Beyond the additional experiments provided in the rebuttal, this is also supported by the results in Table 4, where the state-action-level update variant consistently underperforms PAWS, confirming that single state-action updates are not a good choice in preference-based RL.
> > >
> > > Q3: We appreciate the reviewer raising the practical aspect of tuning. To clarify, we do not claim that $n_{eff}$ is insensitive to data size, but rather that it is more intuitive to set than $\epsilon$. Importantly, the results in Table 1 were obtained using the same hyperparameter value across all 10 tasks and across both the 50- and 500-preference settings, yet PAWS performs on par with or better than the baselines. This demonstrates that a single default setting is already robust in practice, with no per-task or per-dataset-size tuning required. The fact that task-specific tuning could yield further gains is an additional benefit, not a drawback.

---

### Decision · Program_Chairs · 2026-04-30

**Decision:**

Accept (regular)

**Comment:**

The central point in the paper is the mismatch between preferences expressed over segments, vs. per-step signals used during policy optimization. The proposed approach tries to unify both using an advantage function over trajectories.

The reviews highlight that the approach seems simple but sensible. The author rebuttal also points out the connection to other work that argues for reasoning about advantage vs. rewards, so there is good justification. The main criticism is in the potentially limited novelty in the light of Peters et al. (2010). The diagnosis of the problem, however, bears merit, even if the theoretical underpinnings are somewhat studied. Another discussion point raised was the rigor of the human studies. I agree with the reviewer that poor practice in previously published work does not justify following a known flawed methodology, but the author rebuttal did include additional non-author participant data.

Overall, the paper with the revisions based on the rebuttal appear to be a reasonable contribution for ICML. The final revision should be revised to include the newly collected data, and to incorporate the discussions.